# Thermal Behavior and Smoke Suppression of Polyamide 6,6 Fabric Treated with ALD-ZnO and DOPO-Based Silane

**DOI:** 10.3390/ma18133195

**Published:** 2025-07-07

**Authors:** Wael Ali, Raphael Otto, Ana Raquel Lema Jimenez, Sebastian Lehmann, Eui-Young Shin, Ying Feng, Milijana Jovic, Sabyasachi Gaan, Jochen S. Gutmann, Kornelius Nielsch, Amin Bahrami, Thomas Mayer-Gall

**Affiliations:** 1Deutsches Textilforschungszentrum Nord-West gGmbH, Adlerstr. 1, 47798 Krefeld, Germany; 2Institute of Physical Chemistry and Center for Nanointegration (CENIDE), University of Duisburg-Essen, Universitätsstraße 2, 45117 Essen, Germany; 3Institute for Metallic Materials, Leibniz Institute for Solid State and Materials Research, 01069 Dresden, Germany; 4Additives and Chemistry, Advanced Fibers Laboratory, Empa, Lerchenfeldstrasse 5, 9014 St. Gallen, Switzerland

**Keywords:** polyamide (PA6,6), flame retardant, DOPO, atomic layer deposition, ZnO

## Abstract

Polyamide 6,6 (PA6,6) fabrics are widely used in textiles due to their high mechanical strength and chemical stability. However, their inherent flammability and melting behavior under fire pose significant safety challenges. In this study, a dual-layer flame-retardant system was developed by integrating atomic layer deposition (ALD) of ZnO with a phosphorus–silane-based flame retardant (DOPO-ETES). ALD allowed precise control of ZnO layer thickness (50, 84, and 199 nm), ensuring uniform coating. Thermal analysis (TGA) and microscale combustion calorimetry (MCC) revealed that ZnO altered the degradation pathway of PA6,6 through catalytic effects, promoting char formation and reducing heat release. The combination of ZnO and DOPO-ETES resulted in further reductions in heat release rates. However, direct flame tests showed that self-extinguishing behavior was not achieved, emphasizing the limitations related to the melting of PA6,6. TG-IR and cone calorimetry confirmed that ZnO coatings suppressed the release of smoke-related volatiles and incomplete combustion products. These findings highlight the potential of combining metal-based catalytic flame retardants like ZnO with phosphorus-based coatings to improve flame retardancy while addressing the specific challenges of polyamide textiles. This approach may also be adapted to other fabric types and integrated with additional flame retardants, broadening its relevance for textile applications.

## 1. Introduction

Aliphatic polyamides, such as Nylon (PA6,6) and Perlon (PA6), are emerging as essential materials in the world of manufactured textiles not only because of their cost-effectiveness but also due to their distinctive properties such as high mechanical strength, high chemical stability, and resistance against both shrinkage and abrasion [1]. Despite these advantages, the fire performance of PA6 or PA6,6 fabrics remains unsatisfactory, characterized by significant dripping, low limiting oxygen index (LOI) value (20–22%) [1,2], and their lack of flame-retardant efficacy, which is likely due to the inherently aliphatic nature of polyamides [2]. Therefore, various research studies have been conducted to improve the flame retardancy of aliphatic PA textiles [2,3,4,5,6,7,8]. Beyond ignition resistance, smoke generation is a critical aspect of fire safety, particularly for materials used in enclosed or populated environments.

The prohibition and limitation of halogenated flame retardants in the past two decades have spurred researchers and manufacturers to explore diverse alternatives. This shift is driven by the imperative to align with governmental regulations and laws addressing concerns related to human safety, environmental impact, and sustainability.

In this context, metal-based flame retardants, such as metal oxides and hydroxides and metal organic frameworks (MOFs), have gained attention as eco-friendly options [9,10,11,12]. These materials are capable of delivering effective smoke suppression, flame retardancy, and reducing the production of toxic gases [13,14]. The mode of action of metal oxides involves a thermal shielding effect provided by the physical barrier they form, which also enhances char integrity and production. Similarly, MOFs combine catalytic effects and physical barrier properties, promoting char formation, reducing toxic gas emissions, and improving compatibility with polymer matrices. Recent studies have demonstrated their potential to significantly enhance fire resistance in polymeric materials by lowering heat release rates (HRRs) and improving limiting oxygen index (LOI) values [11,12,15]. For textile application, zinc oxide (ZnO) offers excellent properties, including wear resistance, antimicrobial activity, self-cleaning capabilities, and flame retardancy [14,16,17,18,19]. Additionally, ZnO exhibits significant catalytic activity due to its abundant Lewis acid and base sites, enabling diverse chemical transformations. Its catalytic properties have been effectively utilized in fields such as biomass conversion, where ZnO acts as a bifunctional catalyst, facilitating acid–base coordination reactions [20]. Furthermore, the low cost and the ability of ZnO to stabilize intermediates and modify reaction pathways make it a highly attractive material, not only for enhancing flame retardancy in textiles but also for advancing chemical processes involving synthetic and natural polymers.

Smoke suppression in polymers, textiles, and composite flame-retardant systems has garnered increasing attention due to its critical role in fire safety, particularly in reducing toxic gas emissions and minimizing visibility hazards during combustion. Various inorganic and organic additives, such as layered double hydroxides (LDHs) [21], montmorillonite clays [22,23], and low-melting oxide glasses [24], and boron compounds [25,26,27] have been shown to reduce smoke production by promoting char formation, acting as physical barriers, and stabilizing thermal decomposition pathways. In polyamide systems, additives like zinc stannate [28] offer additional advantages by enhancing charring and reducing the release of toxic precursors such as isocyanates. A recent study investigated the performance of several smoke suppressants, including low-melting oxide glass, melem, spherical silica, sepiolite, melamine polyphosphate (MPP), and boehmite, used in combination with aluminum diethylphosphinate, a common phosphorus-based flame retardant for PA6,6 that operates through both gas-phase and condensed-phase mechanisms [29]. The results revealed that all investigated additives enhanced the formation of a protective char layer and acted as effective adjuvants. Among them, silica and melem showed the most substantial performance under forced flaming conditions, with 10 wt% melem reducing total smoke production by 41%. This effect is attributed to melem’s ability to shift the flame-retardant mechanism of aluminum diethylphosphinate from predominantly gas-phase to more condensed-phase activity, thereby enhancing char formation and reducing smoke toxicity.

ZnO can be incorporated into textile substrates using various techniques, such as electrodeposition, pulsed laser deposition (PLD), sol–gel, and atomic layer deposition (ALD), among others [13]. Although ALD of oxides, particularly ZnO, has been widely studied on fabrics, few studies have focused on using ALD as a versatile tool to improve the flame retardancy of fabrics [30,31,32,33]. A study reported the use of atmospheric-pressure spatial atomic layer deposition to coat synthetic polypropylene and cotton fabrics with ZnO, achieving conformal and uniform coatings that enhance UV protection and hydrophobicity. The robust ZnO coatings, which maintained integrity after various durability tests, show promise for the industrialization of functional textiles [13]. In another study by O’Brien et al., an ALD process was also employed to coat wool fabric with a fire-resistant metal oxide (Al_2_O_3_) [34]. The method resulted in treated fabrics that visually resembled untreated ones and exhibited minimal weight gain (specific quantitative data was not provided in the original work). The treated fabrics demonstrated rapid flame extinguishing, as reported by the authors, although visual evidence such as images post-flame test or SEM images were not included. The study noted that the fabrics retained their original properties, including flexibility, breathability, texture, and color. However, it is important to note that wool is inherently flame-retardant due to its high nitrogen and sulfur content and its ability to form a char barrier, which raises questions about the incremental benefit of the ALD treatment. This highlights the need for comparative studies with less flame-retardant substrates to better assess the effectiveness of the ALD coating.

In a recent work, ultrathin TiO_2_, Al_2_O_3_, and ZnO nanofilms were deposited on silk fabric using ALD, significantly enhancing its flame retardancy while preserving its intrinsic comfort and durability. This innovative approach demonstrated superior performance in self-extinguishing flames and maintaining fabric integrity compared to traditional coating methods. However, ZnO is not widely used as a potent flame retardant due to its inability to chemically interact with the degradation process [33]. A closer evaluation of the study reveals several limitations. While the ZnO coatings increased the LOI by 4% compared to untreated samples, which is noteworthy, the flame test images indicate a clear burn-through in the fabric, suggesting that the ZnO coatings, even at higher deposition cycles, do not completely prevent fire damage. Although the study mentions char formation with ZnO coatings, no precise add-on percentages for ZnO coatings for 2000 cycles was provided. Additionally, the study also provides images from cone calorimetry tests, but no associated quantitative data or parameters related to this measurement are included, making it difficult to evaluate the material’s full flame retardancy performance.

The flame-retardant efficacy of ZnO could be significantly enhanced by combining it with reactive flame-retardant materials such as phosphorus- or nitrogen-based compounds. Particularly promising is the combination with gas-phase active species like 9,10-dihydro-9-oxa-10-phosphaphenanthrene-10-oxide (DOPO). For instance, Vasiljević et al. found that DOPO derivatives exhibit excellent flame retardancy in polyamide [35]. The finishing of DOPO derivatives on PA fabric can be achieved through UV grafting techniques or the sol–gel method using DOPO-functionalized silanes [36,37,38,39].

Especially noteworthy are the sol–gel processes, as the application of functionalized silanes enhances the condensed-phase action of DOPO, resulting in substantially improved flame retardancy [40,41]. Sehic et al. reported a synergistic effect between DOPO-based silane and tetraethoxysilane (TEOS) on the flammability of PA6, highlighting the importance of surface treatments in the development of fire-resistant materials [39]. These findings underscore the potential of DOPO derivatives as effective flame retardants for PA6,6, motivating further investigation into their application and optimization. In this context, we hypothesize that implementing a supporting layer of ZnO, followed by a DOPO-based silane derivative (DOPO-ETES) coating, could achieve comparable results for PA6,6 fabric.

## 2. Materials and Methods

### 2.1. Materials

Commercial-woven Nylon fabric (PA6,6) was purchased from WFK-Testgewebe in Brüggen, Germany. The fabric specifications are as follows: fineness [dtex] = 31.5/30, surface weight [g/m^2^] = 75, thread density warp/weft [threads/dm] = 320/310. The flame retardant DOPO-ETES was donated by abcr GmbH (Karlsruhe, Germany). Ethanol (EtOH, 99.8%) was procured from Fisher Scientific GmbH (Schwerte, Germany).

### 2.2. ZnO Coating by ALD

The ZnO thin film was deposited in a thermal ALD reactor (Veeco Savannah S200, Plainview, NY, USA) at deposition temperatures of 100 °C. Diethylzinc (DEZ, Strem, min. 95%) and deionized H_2_O as an oxidizer were pulsed into the reactor in a sequence, using high-purity nitrogen (99.999%) as the carrier and purge gas at 20 sccm. The optimized pulse and purge times for one ALD deposition cycle DEZ/purge/H_2_O/Purge were 0.06/30/0.06/30 s. A growth per cycle (GPC) of ~1.5 Å/cycle at 100 °C was obtained for ZnO grown on Si with a native oxide wafer, as standard reference. Five Si wafers positioned at various locations within the ALD chamber were used to assess uniformity and measure the coating thickness.

### 2.3. Sol–Gel Coating Protocol

Sol–gel coating on ALD-ZnO polyamide fabric was carried out with the following protocol. The sol solutions were initially prepared by dissolving DOPO-ETES silane precursor into a mixture of water and ethanol (*v*/*v*, 50:50) at a concentration of 25 wt%. The pH of the sol solutions was adjusted to 4–5, and the mixture was stirred for 72 h before the coating process.

The sol solution was then applied to the fabric samples using the pad–dry–cure method. This involved full immersion at room temperature for 15 min, followed by a wet pick-up, drying, and curing at 110 °C for 20 min, utilizing a standard laboratory padding machine and an oven. Subsequently, the samples were rinsed with tap water for 10 min. to eliminate excess materials, and the washed fabric was dried under ambient conditions.

To assess the loading of flame retardants, five circular fabric specimens (3 cm in diameter) were cut and weighed before and after treatment with the sol solution. The samples were left for 24 h in a climate-controlled room under standard atmospheric conditions prior to each weighing. The mass add-on values (%) were then calculated as the average of these five replicates.

### 2.4. Measurement and Characterization

The growth per cycle (GPC) was measured on a Si reference wafer by using an ellipsometer (Sentech Instrument GmbH, Berlin, Germany).

Fourier transform infrared spectroscopy (FTIR) measurements were conducted using an IR Prestige-21 instrument (Shimadzu Deutschland GmbH, Duisburg, Germany) in attenuated total reflection (ATR) mode, with an average of 100 scans per sample and a resolution of 2 cm^−1^. The ATR was conducted using a golden gate equipped with a diamond crystal (Specac Ltd., Orpington, UK).

Thermogravimetric analyses (TGA) were conducted using a Discovery TGA 55 instrument (TA Instruments, Hüllhorst, Germany) over a temperature range of 40 to 800 °C. Approximately 10 mg of the sample, previously dried for 24 h at 60 °C, was positioned in a platinum crucible. The temperature was initially held at 100 °C for 5 min to remove adsorbed water, followed by heating at a rate of 20 °C/min under nitrogen atmospheres, with a gas flow rate of 90 mL/min.

In parallel with thermogravimetric measurements, the volatile degradation products were identified through TGA-FTIR analysis. For this purpose, the TGA system was interfaced with the previously described FTIR spectrometer via a heated transfer line and gas cell, both maintained at 240 °C to prevent condensation of volatiles. Infrared spectra were acquired continuously during the thermal decomposition, with each spectrum obtained as the average of four scans, recorded at a resolution of 4 cm^–1^ over the spectral range of 600 to 4000 cm^−1^.

Microscale combustion calorimeter (MCC) measurements were carried out using a Fire Testing Technology Ltd. instrument (East Grinstead, UK) following ASTM D 7309 Method A. In each experimental run, the sample underwent pyrolysis up to 750 °C with a heating rate of 1 °C/s in a nitrogen stream flowing at a rate of 80 mL/min. Subsequently, the volatile thermal degradation products obtained were combusted at 900 °C in a combustion chamber upon mixing with a nitrogen–oxygen gas stream (80/20) at a flow rate of 20 mL/min. The recorded data were analyzed using Origin 2018b software (OriginLab Cooperation, Northampton, MA, USA). Each sample was tested in triplicate, and the values reported are the average of these three measurements.

The fire growth capacity (FGC) was calculated using the following Equation (1):(1)$$FGC=\left(\frac{THR}{T_{95\%}-T_{5\%}}\right)\left(\frac{T_{95\%}-298K}{T_{5\%}-298K}\right)$$

Here, THR stands for total heat release, while *T*_5%_ and *T*_95%_ denote the temperatures corresponding to the ignition and burning points, respectively, which are determined based on the 5% and 95% points of the integrated heat release rate (HRR) curve.

Scanning electron microscopy (SEM) analysis was conducted with an S-3400 N II SEM instrument from Hitachi HighTech Europe GmbH, Mannheim, Germany, operating at an accelerating voltage of 10 kV. To enhance conductivity and imaging quality, the sample surfaces were coated with a thin layer of gold using a Quorum Emitech K500X sputter coater (Ashford, Kent, UK) in a vacuum for 4 min.

The surface frictional characteristics of pristine and ZnO-coated PA6,6 fabrics were also characterized using scanning probe microscopy (SPM). Measurements were performed on an Agilent 5500 SPM system (Santa Clara, CA, USA) operated in contact mode, equipped with silicon probes (Nanosensors) featuring a nominal force constant of 0.01–1.87 N/m and typical dimensions of 225 μm in length, 48 μm in width, and 10–15 μm tip height. Scans were conducted at a resolution of 1024 data points per line with a scan speed of 1 line per second and a scan area of 5 µm × 5 µm. Post-processing and image analysis were performed using Gwyddion software (version 2.57).

To assess the size distribution of the sol nanoparticles, dynamic light scattering (DLS) analysis was conducted using a Zetasizer 1000 system (Malvern Panalytical Ltd., Malvern, UK). The sample was diluted 1000-fold to reduce the likelihood of agglomeration. The measurements were taken at a 90° angle and maintained at a temperature of 25 °C.

X-ray photoelectron spectroscopy (XPS) was employed to assess the elemental composition of the pristine and ZnO-coated PA6,6 fabrics. Measurements were carried out using a PHI 5000 VersaProbe II system (Physical Electronics, Chanhassen, MN, USA) equipped with a monochromatic Al Kα source (photon energy 1486.6 eV) and a 100 µm beam diameter. Survey spectra were acquired at a photoelectron take-off angle of 45° under charge neutralization conditions. CasaXPS software (version 2.3.26) was used for data processing [42].

Small- and wide-angle X-ray scattering (SAXS/WAXS) measurements were conducted using a Xenocs XEUSS system equipped with a Cu Kα radiation source (λ = 1.54 Å) and an Eiger 2R 1M two-dimensional pixel detector. Data were collected at two sample-to-detector distances: 43 mm for WAXS and 800 mm for SAXS. Line cuts were extracted from the two-dimensional (2D) SAXS/WAXS patterns to obtain one-dimensional (1D) scattering profiles. Data analysis was performed using the software Xenocs XSACT (Version 2.7, 2022).

Flame tests were conducted according to EN ISO 15025:2016 [43] (Protective clothing—Protection against heat and flame—Method of test for limited flame spread, 10 s ignition) utilizing the Gester flammability tester (Model GT-C35B, Gester International Co., Ltd., Quanzhou, China). The tests implemented the surface ignition procedure, with a flame height set at 25 mm. Compliance with passing requirements was evaluated based on ISO 11611:2024 standards [44]. The textile samples, with a modified size of 10.5 cm × 6.0 cm, were used for the tests. Propane was used as the burning gas. Each sample was tested in triplicate. To quantify the flammability behavior, after-flame time (AFT) was recorded in seconds. Additionally, burned area (BA) was calculated as a percentage of the total sample area from digital photographs using the ImageJ free software (version 1.53t), by thresholding and segmenting the damaged regions.

Cone calorimetry measurements were conducted in accordance with ISO 5660 [45] using a heat flux of 35 kW/m^2^. Samples were mounted horizontally in a holder, wrapped in aluminum foil, and secured with a metallic grid. The exposed surface area was 88.4 cm^2^, with the sample positioned 25 mm below the cone heater. Air was supplied at a rate of 24 L/s, and the ambient conditions during testing were 22.5 °C with 28.2% relative humidity. Data were collected at 5 s intervals over a total duration of 300 s. The HRR and gas emissions (CO and CO_2_) were measured in real time, from which parameters such as THR, time to ignition (TTI), and maximum average rate of heat emission (MARHE) were calculated. To quantify smoke release, the setup also directly measured the total smoke production (TSP) and the CO/CO_2_ ratio, providing insight into the completeness of combustion and smoke density. Each measurement was performed in triplicate, and average values were reported.

## 3. Results

PA6,6 fabrics are particularly difficult to render flame-retardant due to their tendency to melt and drip under high temperatures, leading to rapid flame propagation. In this study, we evaluate the effectiveness of a dual-layer flame-retardant system combining a ZnO underlayer and a DOPO-based nanosol topcoat in enhancing the flame retardancy of PA6,6 fabrics. The ZnO underlayer not only acts as a physical barrier but is also expected to provide a reactive surface for strong bonding with silane molecules from the DOPO-based nanosol, resulting in a cohesive and uniform coating [46,47,48,49,50,51]. This combination is anticipated to utilize the physical barrier properties of ZnO and SiO_2_, as well as the chemical flame-retardant properties of DOPO, to address the inherent challenges of flame-retarding PA6,6.

### 3.1. PA6,6 Finishing and Characterization

The dual-layer treatment of PA6,6 fabric with ZnO and DOPO-based silane was systematically performed as illustrated in Figure 1A.

The ZnO layer was deposited using ALD. To evaluate the effect of ZnO thickness on the flame-retardant properties, three different ZnO thicknesses were studied, as shown in Figure 2. These variations in ZnO thickness, achieved through adjustments in the number of ALD cycles, result in noticeable differences in the fabric’s appearance and morphology (Figure 2a). Coated fabrics exhibit thicker ZnO layers of 50 nm, 84 nm, and 199 nm at deposition cycles of 300, 600 and 1200, respectively, corresponding to weight gains of approximately 3%, 5%, and 9% (Table 1). For clarity, the samples are denoted as ZnO-50, ZnO-84, and ZnO-199, where the numerical value refers to the approximate ZnO layer thickness (in nm) as determined from reference wafers.

The ALD process enabled precise, layer-by-layer deposition of ZnO, forming conformal coatings on the PA6,6 fabric while retaining the underlying textile morphology. As shown in Figure 2a, the ZnO-coated samples exhibit a progressively more pronounced yellowish tint with increasing layer thickness, reflecting the accumulation of ZnO. SEM images (Figure 2b) demonstrate that the fibrous weave structure of the fabric remains intact across all coatings. The ZnO layers conform closely to the individual fibers and preserve the textile’s microstructure, indicating that the ALD process does not disrupt the fabric’s physical integrity. With increasing thickness, a more uniform distribution of ZnO is visible, with fewer visible voids compared to the thinner coatings, although without significant alteration to the macroscopic texture.

Scanning probe microscopy (SPM) images (Figure 2c) provide complementary nanoscale insights. The surface of pristine PA6,6 shows oriented features corresponding to the native fiber texture, whereas the coated samples reveal the presence of ZnO nanoparticles homogeneously distributed across the surface. The friction images highlight the consistent dispersion of the ZnO coating, supporting the conclusion that the ALD process delivers even nanoscale coverage throughout the fabric surface.

XPS survey spectra (Figure 2d) confirm the successful surface modification of PA6,6 through ZnO deposition. In all ZnO-coated samples, characteristic signals for Zn (Zn 2p and Zn LMM) and O (O 1s) are clearly detected, verifying the presence of ZnO on the fabric surface. Interestingly, an increase in the intensity of the C 1s signal is observed with higher ALD cycle numbers. This rise is not attributed to the polymer substrate, which is constant across samples and as the N 1s signal disappears after ZnO coating, but is more likely due to residual carbon-containing species from the ALD precursor, diethylzinc (DEZ). Incomplete removal of these organic ligands during low-temperature deposition may lead to their accumulation on the surface, particularly at higher deposition cycles.

This controlled deposition process highlights the versatility of ALD in fine-tuning the thickness and uniformity of ZnO coatings, enabling a balance between the functional properties and the fabric’s flexibility and appearance.

In addition to SPM and XPS analyses, small- and wide-angle X-ray scattering (SAXS and WAXS) measurements were conducted to assess the structural characteristics of the ZnO-coated PA6,6 fabrics (Figure 3 and Appendix A). These measurements not only confirm the presence of ZnO nanoparticles on the fabric surface but also demonstrate an increase in ZnO crystallinity with greater deposition thickness. The SAXS data (Appendix A) show a systematic increase in scattering intensity with increasing ZnO layer thickness, indicating the formation of ZnO-based nanostructures. Notably, the broad peak centered around q ≈ 0.80 Å^−1^, present in pristine PA6,6, becomes less prominent after coating, suggesting alterations in the local surface morphology and polymer interfacial ordering due to the inorganic layer.

The 2D WAXS patterns (Figure 3, left) illustrate the structural differences between pristine and ZnO-coated PA6,6 fabrics. The pristine sample shows a diffuse halo typical of semi-amorphous polymers, indicating a lack of long-range order. Upon ZnO deposition, distinct diffraction rings begin to appear and become more defined with increasing layer thickness. This evolution reflects the formation of crystalline ZnO domains and suggests increasing structural order within the coating. To quantify this structural development, 1D WAXS line profiles were obtained from radial integration of the 2D patterns (Figure 3, right). The profiles exhibit a series of sharp peaks whose intensities increase systematically with ZnO thickness. Specifically, peaks located at q = 2.34 Å^−1^, 2.52 Å^−1^, 2.65 Å^−1^, and 3.39 Å^−1^ correspond to the (100), (002), (101), and (102) crystallographic planes of the wurtzite ZnO structure, confirming the presence of a well-defined crystalline phase [52]. In contrast, the pristine PA6,6 displays only a broad scattering band in the q-range of 0.8–2.0 Å^−1^, consistent with its disordered nature. Notably, no shift or modification of the characteristic PA6,6 scattering features was observed upon coating, indicating that the polymer’s semi-crystalline structure remains unaffected by the ALD deposition of ZnO.

Following the ALD process, the application of the DOPO-based silane involved two distinct steps: the preparation of the nanosol via the sol–gel method and its subsequent application to the fabric using the pad–dry–cure process. The sol–gel process, as represented in Figure 1B, involves the hydrolysis and condensation of DOPO-ETES precursors in an aqueous solution. This process resulted in the formation of a stable nanosol with uniform particles. DLS analysis, shown in Figure 1C, confirmed the nanosol’s particle size distribution, with an average diameter of approximately 350 nm. The pad–dry–cure process was then employed to uniformly apply the prepared nanosol onto the ZnO-coated PA6,6 fabric, as shown in Figure 4a.

The mass add-on values (weight gain) after the treatment of both PA6,6 and PA6,6-ZnO fabrics with DOPO-ETES were maintained at nearly the same levels (~15 wt%), as presented in Table 2, to ensure good comparability between the samples. The fabric treated directly with DOPO-ETES is referred to as PA6,6-DOPO, while fabrics coated sequentially with ZnO and DOPO-ETES are referred to as ZnO–DOPO-treated, indicating the dual-layer configuration, where the numerical value (e.g., ZnO-199-DOPO) refers to the approximate ZnO layer thickness in nm. The SEM images in Figure 4a illustrate the surface morphology of PA6,6 fabrics coated with DOPO-based silane alone (PA6,6-DOPO) and in combination with ZnO layers of varying thicknesses. The DOPO coating can be observed as a uniform and continuous layer, adhering to the underlying ZnO-coated fabric. Additionally, the coating is visible between the fibers, contributing to more complete surface coverage. The presence of ZnO clearly enhances the uniformity and distribution of the silane layer compared to pristine PA6,6, with no significant difference observed between the coatings for the different ZnO thicknesses.

Figure 4b displays ATR-IR spectra of pristine PA6,6, PA6,6 coated with DOPO-ETES and of PA6,6 fabrics coated with ZnO (199 nm) with and without the DOPO-based nanosol. For pristine PA6,6, characteristic peaks are observed at 3292 cm^−1^ (N–H stretching), 2932 cm^−1^ and 2854 cm^−1^ (C–H stretching), as well as 1631 cm^−1^ (C=O stretching in amide groups) and 1533 cm^−1^ (N–H bending combined with C–N stretching). These peaks are consistently present in all samples, indicating that the bulk signals of the PA6,6 substrate dominate and mask any subtle contributions from the ZnO layer. This is due to the relatively thin ZnO coating, whose signals are not strong enough to appear distinctly in the spectra. Upon coating with DOPO-based silane, new peaks emerge, confirming the successful surface treatment of the fabrics. A characteristic peak at 1199 cm^−1^ is attributed to the stretching vibrations of P=O, while the peak at 914 cm^−1^ corresponds to the stretching vibrations of P–O–Ph, further verifying the introduction of phosphorus-containing structures from the DOPO-based silane. Additionally, the peak at 753 cm^−1^ is attributed to aromatic hydrogen vibrations, consistent with the aromatic structure of DOPO.

Figure 4c confirms the successful surface modification of PA6,6 with the DOPO-based silane. Compared to the pristine fabric, which shows signals for C 1s, O 1s, and N 1s, the coated samples display clear peaks corresponding to P 2p and Si 2p, verifying the presence of phosphorus- and silane-containing functionalities. These signals are evident in both the PA6,6-DOPO and PA6,6-ZnO-199-DOPO samples, supporting the effective deposition of the DOPO-based silane. A small ZnO-related signal is also visible in the latter, consistent with the underlying ZnO layer. The atomic percentages of P and Si obtained from XPS are slightly higher than the corresponding theoretical bulk values (see Table 2), which can be attributed to the surface-sensitive nature of the technique. Nevertheless, the consistent elemental ratios across samples support the presence of a relatively uniform and homogeneous coating.

### 3.2. Thermal Decomposition and Combustion Behavior

#### 3.2.1. TG Test

The thermal stability of pristine PA6,6 and ZnO-coated PA6,6 fabrics with varying layer thicknesses was evaluated using TGA and differential thermogravimetric (DTG) curves, as shown in Figure 5, and a summary of the TGA results is provided in Table 3. The TG curves show that pristine PA6,6 exhibits a single-stage decomposition process, with the temperature for 5% weight loss (T_5%_) at 401 °C and a maximum decomposition temperature (T_max_) at 471 °C. Additionally, a shoulder peak is observed around 400 °C, which is attributed to the onset of the degradation process. For ZnO-coated samples, a slight shift in both T_5%_ and T_max_ to lower temperatures is observed, with T_5%_ ranging from 387 °C to 391 °C and T_max_ ranging from 443 °C to 445 °C. However, no significant differences in these temperatures are observed with increasing ZnO thickness, suggesting that the presence of ZnO influences the decomposition process in a similar manner across all layer thicknesses.

The residue at 700 °C increases gradually as the ZnO thickness, and hence the add-on value of ZnO, increases. Pristine PA6,6 leaves almost no residual weight (0.98%), while the residue for ZnO-coated samples increases to 3.8% for PA6,6-ZnO-50 and reaches 10.4% for PA6,6-ZnO-199. This trend suggests that ZnO plays an active role in influencing the decomposition of PA6,6, beyond its inert, non-combustible nature. ZnO, as an amphoteric material, can act as an acid–base catalyst, potentially catalyzing the cleavage of amide bonds in the polymer chain under thermal or oxidative conditions [20]. This catalytic activity accelerates the breakdown of PA6,6, potentially contributing to the earlier release of volatile degradation products and the formation of stable residue during the decomposition process.

The DTG curves further support this, showing a slight reduction in the maximum degradation rate for ZnO-coated fabrics compared to pristine PA6,6. This behavior indicates that the ZnO layer may alter the degradation pathway of PA6,6 through its catalytic properties. The catalytic activity of ZnO likely facilitates the breakdown of polymer chains into intermediate products, promoting the formation of stable residues. This residue formation enhances the thermal stability of the material in the later stages of decomposition. This catalytic role of ZnO supports the conclusion that ZnO coatings improve both the thermal decomposition behavior and flame retardancy of PA6,6 fabrics.

The impact of DOPO-based silane coatings combined with ZnO layers of varying thicknesses on the decomposition behavior of PA6,6 fabrics was also investigated, as illustrated in Figure 6. These results provide insights into how the dual-layer coating system alters the thermal behavior of the fabrics compared to pristine PA6,6.

The TGA curves (Figure 6, top) show that the addition of the DOPO-nanosols coating alone (PA-DOPO) leads to a slightly lower T_5%_ (391 °C) and T_max_ (449 °C) compared to pristine PA6,6, indicating that the DOPO-ETES layer decomposes just before the onset of PA6,6 decomposition. Similar findings have been previously reported [37,39], attributing this behavior to the early degradation of DOPO.

The DTG curves (Figure 6, bottom) reveal that the maximum degradation rate of PA6,6-DOPO is slightly higher than that of pristine PA6,6, suggesting that the DOPO-based coating accelerates the degradation process in the main decomposition stage.

This behavior is likely due to the fact that the decomposition of PA6,6 and the DOPO-based silane occur independently rather than synergistically. The degradation of DOPO-based silane is likely initiated at lower temperatures due to its phosphorus content, which promotes early char formation and releases gaseous products. This earlier decomposition can overlap with the degradation of PA6,6, increasing the overall release of volatiles during the main decomposition stage, thereby accelerating the degradation process and increasing the maximum degradation rate.

However, when the two coatings are combined (PA-ZnO-DOPO), the maximum degradation rate decreases again, reaching values equal to or even slightly lower than those of PA6,6-ZnO samples. This behavior suggests that the combined system not only incorporates the effects of both ZnO and DOPO-based silane but also probably exhibits a synergistic interaction. ZnO likely catalyzes the degradation of PA6,6 into intermediates that facilitate char formation, while the DOPO-based silane enhances this effect, in addition to its possible gas-phase activity, resulting in an interaction that mitigates the accelerating effect observed in PA6,6-DOPO and improves the overall degradation-slowing behavior.

The reduction in maximum degradation rate for PA6,6-ZnO-DOPO samples is consistent regardless of ZnO thickness, indicating that the presence of DOPO stabilizes the degradation process in conjunction with ZnO, unlike PA6,6-ZnO with a thickness of 50 nm, where the reduction in degradation rate is less pronounced.

Similarly, the T_max_ of PA6,6-ZnO-DOPO samples increases slightly compared to PA6,6-DOPO, regardless of ZnO thickness. For instance, T_max_ for PA6,6-ZnO-50-DOPO and PA6,6-ZnO-199-DOPO is 457 °C and 459 °C, respectively, compared to 449 °C for PA6,6-DOPO and 443–445 °C for PA-ZnO samples. This slight increase suggests that the DOPO-based silane coating interacts with the ZnO layer, potentially moderating the catalytic activity of ZnO by partially covering its active sites or altering the degradation pathway. This interaction, combined with the thermal barrier effects of both coatings, results in a more stable decomposition process.

The residual mass at 700 °C further supports the interaction between ZnO and DOPO-ETES, with PA6,6-ZnO-DOPO samples showing the highest residues due to the combined contribution of ZnO and the char-promoting properties of the DOPO-ETES coating. Figure 7 illustrates the residues of the studied samples after the TG test, highlighting the individual contributions of ZnO and DOPO-based silane and the PA6,6 substrate to the total residue. It is evident that the residue from the DOPO-based silane layer is not solely from the silane inorganic remnants, as the maximum weight percentage of SiO_2_ (calculated from Table 2) is around 3% (see also Table 3). This suggests that the DOPO-based silane coating contributes to char formation beyond its inorganic SiO_2_ content, likely due to the phosphorus content, which promotes char formation during thermal decomposition.

The figure also reveals that the ZnO layer significantly contributes to the total residue, with thicker ZnO coatings leading to higher residue percentages. This aligns with the role of ZnO as a non-combustible material that remains stable at high temperatures. The synergistic effect of ZnO and DOPO-based silane is reflected in the increased overall residue for PA6,6-ZnO-DOPO samples, indicating that the dual-layer coating effectively enhances the char residue and thermal stability of the treated PA6,6 fabrics.

Moreover, the theoretical residue values in Table 3 were calculated to evaluate the contribution of ZnO and SiO_2_ to the overall residue formation. For samples treated with ZnO alone, the theoretical residue was calculated based on the residue of pristine PA6,6 at 700 °C and the weight percentage of ZnO derived from Table 1. For samples treated with both ZnO and DOPO-based silane, the theoretical residue includes the calculated contribution of SiO_2_ based on Table 2. The observed residue values are generally higher than the theoretical values, particularly for samples treated with DOPO-based silane. This indicates that DOPO promotes additional residue formation, likely through char-promoting mechanisms, in combination with ZnO. These results highlight the potential of the dual-layer system to enhance thermal stability, though further studies are needed to confirm the synergistic interactions between ZnO and DOPO.

#### 3.2.2. MCC Test

The microscale combustion calorimeter (MCC) is an effective tool for evaluating the combustion behavior of flame retardant-treated materials. This method allows for rapid screening of combustion parameters such as the peak heat release rate (pHRR), total heat release (THR), heat release capacity (HRC), and fire growth capacity (FGC), which are critical for assessing flame-retardant performance. Different studies have reported a correlation between MCC results and larger-scale tests, such as cone calorimetry, particularly for pure polymers [53,54,55]. However, it is worth noting that MCC might have limitations in fully predicting large-scale fire performance for flame-retardant materials, as certain mechanisms, such as barrier effects and flame inhibition, may not be entirely reflected in MCC results. Nonetheless, MCC remains a powerful and efficient tool for analyzing combustion properties and provides valuable insights that can guide material development. Table 4 summarizes the MCC data, while Figure 8 and Figure 9 illustrate the HRR and THR curves of pristine PA6,6 and coated PA6,6 fabrics with DOPO-ETES for different ZnO layer thicknesses.

The HRR curves (Figure 8, left) reveal distinct differences between pristine PA6,6 and ZnO-coated fabrics. Pristine PA6,6 exhibits the highest pHRR at 547 W/g, with the corresponding T_pHRR_ at 475 °C. This high value reflects the rapid combustion of PA6,6 and its poor intrinsic flame retardancy. A shoulder peak is also observed around 400 °C, which corresponds to the early stages of decomposition. The addition of ZnO layers slightly reduces the pHRR for all coated samples, with values decreasing as the ZnO thickness increases. Similar to what was observed in the TGA data, the T_pHRR_ also shifts to lower temperatures compared with pristine PA6,6, without any noticeable difference between the different ZnO thicknesses.

The THR curves (Figure 8, right) align closely with the TGA data, showing a reduction in total heat release for ZnO-coated samples compared to pristine PA6,6. The THR decreases as the ZnO thickness increases (see also Figure 10C). The catalytic effect of ZnO likely facilitates the degradation of PA6,6 into intermediate products that promote char formation and reduce the release of combustible gases. This catalytic activity plays a crucial role in lowering the overall heat release during combustion.

Figure 9 illustrates the HRR (top) and THR (bottom) curves of pristine PA6,6 and PA6,6 fabrics coated with DOPO-ETES and ZnO layers of different thicknesses. The HRR curves highlight that PA6,6-DOPO exhibits a higher peak heat release rate (pHRR) compared to pristine PA6,6, indicating that the phosphorus-containing DOPO-ETES coating accelerates the combustion process during the main decomposition phase. Such behavior has also been reported in the literature for DOPO derivatives with PA6,6. However, when DOPO-ETES is combined with ZnO, the pHRR is significantly reduced compared to both individual treatments, suggesting a possible synergistic effect between the two components. This reduction in HRR reflects the enhanced flame retardancy achieved through the dual-layer system.

An additional peak or shoulder can also be observed at an T_pHRR_ of approximately 425 °C, becoming more prominent as the ZnO thickness increases. This new peak likely arises from the interaction between the ZnO and DOPO-ETES layers, representing a distinct decomposition event. In addition to PA6,6, ZnO may catalyze the breakdown of DOPO-ETES, facilitating the release of phosphorus-containing volatiles at lower temperatures. This catalytic activity, along with the interaction of ZnO with the residual SiO_2_ from DOPO-ETES, stabilizes intermediates and leads to the emergence of this new decomposition peak. Furthermore, the dual-layer system appears to alter the degradation pathway of PA6,6, further contributing to the development of this distinct feature.

The THR curves (Figure 9, bottom) show a consistent reduction in THR for the dual-layer coatings compared to both pristine PA6,6 and the individual treatments (DOPO-ETES or ZnO alone), enhancing the interaction between DOPO-ETES and ZnO. This effect is particularly pronounced with increasing ZnO thickness, indicating that thicker ZnO layers contribute to greater thermal stability and char residue (see also Figure 10C).

To gain insight into the changes in various parameters derived from the MCC data, Figure 9 illustrates the fire growth capacity (FGC, Figure 10A), heat release capacity (HRC, Figure 10B), total heat release (THR, Figure 10C), and residues (Figure 10D) as functions of ZnO layer thickness on PA6,6 fabrics, both without and with the DOPO-based silane coating.

The FGC values for samples treated with ZnO alone remain relatively unchanged across all ZnO thicknesses, except for the PA6,6-ZnO-50 sample, which shows a higher FGC, likely due to incomplete development of the thermal barrier effect of ZnO and a concentrated release of volatiles. In contrast, the addition of DOPO-ETES significantly reduces the FGC across all ZnO thicknesses, reflecting its synergistic interaction with ZnO in reducing flame growth through char formation and gas-phase activity. The HRC values indicate that PA6,6-DOPO has a significantly higher HRC compared to pristine PA6,6 or ZnO-treated PA6,6, highlighting the acceleration of combustion caused by the DOPO-based silane during the main decomposition phase. However, the combination of DOPO-ETES and ZnO results in an exponential decrease in HRC with increasing ZnO thickness, demonstrating the dual-layer system’s ability to suppress heat. The THR values consistently decrease with the application of ZnO and DOPO-ETES. As discussed above, ZnO alone slightly reduces THR due to its non-combustible nature, while the dual-layer system further suppresses THR, enhancing flame retardancy. Similar to the findings in the TGA data, the residue data show a significant increase with both ZnO and DOPO-ETES coatings, with DOPO-treated samples exhibiting higher residues than those treated with ZnO alone.

#### 3.2.3. Cone Calorimeter Test

Cone calorimetry was conducted to evaluate the fire behavior of pristine and coated PA6,6 fabrics under a heat flux of 35 kW/m^2^. Due to the limited available mass (~1 g) of textile specimens, the test deviates from standard conditions typically applied in cone calorimetry, which recommend a sample mass of 20–50 g for accurate combustion analysis. This low fuel load can lead to rapid burnout, unstable ignition, and underrepresentation of heat release values. As such, selected quantitative results (e.g., pHRR, THR, TTI, and MARHE) showed inconsistent trends, e.g., pristine PA6,6 displaying lower peak values than all treated fabrics, and no clear trend was observed among the treated samples and therefore is not discussed here (see Appendix A). Despite these limitations, the data provide meaningful insights into smoke suppression and combustion behavior.

As shown in Figure 11a, the total smoke production (TSP) is significantly reduced in ZnO-coated samples compared to pristine PA6,6, with increasing ZnO layer thickness leading to further reductions. This behavior is attributed to the barrier effect of the ZnO, which limits the release of volatile, smoke-forming species and promotes the retention of residues in the condensed phase. In contrast, DOPO-treated samples, both PA6,6-DOPO and PA6,6-ZnO-199-DOPO, exhibit substantially higher TSP values, suggesting an increased release of condensed-phase decomposition products. This increase is linked to the condensed-phase decomposition of the DOPO-based flame retardant, which is associated with the formation of incompletely oxidized, tar-like products that contribute to visible smoke. The sample with both ZnO and DOPO shows a modest reduction in TSP compared to DOPO alone, indicating that the ZnO sub-layer partially suppresses the smoke-promoting effect of the silane treatment.

The CO/CO_2_ ratio, shown in Figure 11b, offers additional insight into the combustion behavior of the treated fabrics. For the ZnO-coated PA6,6 samples, no significant change in the CO/CO_2_ ratio was observed across the different ZnO thicknesses, suggesting that the ZnO layer, regardless of thickness, does not substantially affect the oxidation pathway of the released volatiles. This indicates that the ZnO coating functions primarily as a physical barrier, limiting fuel release and smoke formation (as reflected in the TSP results), but does not significantly enhance oxidative combustion. In contrast, the DOPO-treated sample exhibited a pronounced increase in CO/CO_2_ ratio, indicating a shift toward incomplete combustion, possibly due to alterations in thermal degradation behavior introduced by the DOPO-based coating. Interestingly, the sample treated with both ZnO-199 and DOPO-ETES showed a lower CO/CO_2_ ratio than DOPO alone, suggesting that the ZnO underlayer partially reduces the tendency toward incomplete combustion induced by DOPO. However, the ratio remains higher than in pristine PA6,6 or ZnO-only samples.

These findings suggest that ZnO and DOPO-based silanes influence combustion behavior through different mechanisms, but their combined effect is not strongly synergistic under the current conditions. It should be noted that in the current layered configuration, the DOPO-based silane is deposited over the ZnO coating and therefore may have limited direct interaction with the polymer surface, which could influence its flame-retardant effectiveness.

### 3.3. TG-IR Analysis

TG-IR analysis was employed to monitor the evolved gases and volatile degradation products of pristine PA6,6 and coated samples. The IR spectra at the maximum degradation temperature (T_max_), as well as the cumulative signal intensities of selected wavenumbers across the full temperature range, are presented in Figure 12, while Appendix A illustrate the temperature-dependent evolution of specific volatiles. At T_max_, characteristic signals of typical PA6,6 degradation products were identified in all samples, most notably hydrocarbons (2935 cm^−1^), CO_2_ (2318 cm^−1^), carbonyl species (1712 cm^−1^), and ammonia (964 cm^−1^).

Coating with ZnO did not notably alter the overall degradation mechanism, as evidenced by the similarity of IR spectral profiles compared to pristine PA6,6 (Figure 12A). However, a reduction in the total emission of combustibles (both for hydrocarbons and carbonyls) with increasing ZnO layer thickness was observed (Figure 12C). This suggests that the ZnO coating functions as a physical barrier, interfering with the release of combustible volatiles and promoting the retention of carbonaceous residues in the condensed phase. This interpretation is consistent with the higher residual char yields observed in TGA and the reduced pHRR and THR values in MCC tests. The barrier effect becomes more pronounced at higher thicknesses, where the restricted transport of volatile species toward the gas phase may enhance secondary reactions, such as radical recombination, promoting tar or char formation.

Upon application of DOPO-ETES alone, a distinct shift in degradation behavior becomes evident. While hydrocarbon emissions remain comparable to the ZnO-199 coating, a substantial decrease in carbonyl signals is evident, alongside a marked increase in ammonia release (Figure 12B,D). This shift indicates that DOPO-ETES modifies the cleavage pathway of the polyamide chain, promoting amide bond scission and NH_3_ evolution, suggesting pronounced condensed-phase activity. Although no distinct gaseous phosphorus-containing species were observed in the IR spectra, it should be noted that characteristic P–O and P=O stretching vibrations, typically expected in the 1250–900 cm^−1^ region, may be masked by overlapping degradation signals from PA6,6. These include strong C–O, C–N, and N–H absorption bands, which dominate this spectral range and may obscure weak phosphorus-related signals. Furthermore, the limited sensitivity of TG-IR for low-concentration volatiles under such conditions cannot fully exclude the possibility of minor gas-phase phosphorus species. Nevertheless, our findings are consistent with literature reports indicating that DOPO derivatives functionalized with silane groups exhibit enhanced condensed-phase activity on PA6,6 fabrics [37]. In contrast, DOPO derivatives lacking silane functionality have been shown to act more prominently in the gas phase. Thus, the absence of observable gas-phase phosphorus signals in this study supports the interpretation that DOPO-ETES acts predominantly in the condensed phase in the ZnO–DOPO system applied to PA6,6 fabrics [36].

Combining ZnO-199 and DOPO-ETES on PA6,6 results in reduced interaction between the DOPO moiety and the polymer surface, likely due to the physical barrier imposed by the ZnO layer. This is supported by the lack of detectable changes in the degradation mechanism, as observed in the IR spectra. The emission of hydrocarbons remains comparable across all treated samples but is clearly lower than in pristine PA6,6, indicating that each treatment (ZnO, DOPO, or their combination) contributes to partial suppression of these high-energy volatiles. In contrast, the carbonyl emissions are significantly reduced in all treated fabrics, with the DOPO-treated fabric showing the most pronounced suppression, followed by the ZnO–DOPO-treated fabric, both showing greater suppression than the ZnO-only sample. Since carbonyl-containing compounds are associated with partially oxidized degradation intermediates, this reduction suggests altered decomposition chemistry in the presence of DOPO.

In summary, the TG-IR data suggest that while ZnO primarily acts as a physical barrier, DOPO-ETES contributes to chemical modification of the decomposition process. The combined treatment leads to an additive effect, particularly in carbonyl suppression, which complements the improved flame retardant and smoke suppression behavior observed in MCC and cone calorimetry.

### 3.4. Flame Test

The direct flame behavior of both pristine and treated PA6,6 samples was difficult to assess due to the thermoplastic nature of the material. Multiple standard test methods, including DIN EN ISO 15025 with surface and bottom edge ignition, DIN 4102-1 (B2) [56], which also uses bottom edge ignition, and the horizontal flame test (DIN 7500) [57], were evaluated. However, consistent flame test performance could not be ensured, even when repeating the same test on identically prepared samples. This variability is due to the pronounced melting and shrinking behavior of PA6,6, which prevents sustained burning or self-extinguishing behavior and undermines the comparability of results. None of the tested samples passed any of the applied flame test standards. Additional trials with increased DOPO-based silane add-on levels up to 25 wt% were also conducted, but the flame-retardant performance remained insufficient to meet the required standards. Nonetheless, as an example, Figure 13 shows the results of the flame test conducted according to EN ISO 15025 with surface ignition and modified sample sizes (see the characterization section).

Pristine PA6,6 exhibited extensive burning and melt-dripping, leaving a large central burned area (BA = 41 ± 20%) with an after-flame time (AFT = 11 ± 7 s). ZnO-coated fabrics showed slightly increased burned areas (e.g., up to 68 ± 25% for ZnO-199) and similar AFTs (12–19 s), but reduced dripping behavior was observed with increasing ZnO thickness. Blackened char edges were visible in higher-thickness samples. In contrast to both pristine and ZnO-only treated fabrics, PA6,6 samples treated with the DOPO-based silane exhibited a complete suppression without after-flame time and significantly reduced burned areas (e.g., 20–35%), along with an absence of dripping. These results suggest that the DOPO-based silane coating contributes to improved flame-retardant behavior, particularly through condensed-phase action and thermal stabilization. However, even the dual-layer ZnO–DOPO treatments failed to fully suppress the inherent melting of PA6,6 and did not achieve self-extinguishing behavior under flame exposure. This melting and shrinking behavior also hindered the accurate determination of the limiting oxygen index (LOI), as sample deformation and melt-dripping, rather than burning, compromised the reliability of LOI measurements for both pristine and treated fabrics.

It should also be noted that in many studies on flame-retardant treatments for PA6,6 fabrics, results from the vertical flame test (UL 94), despite its original design for rigid polymer specimens rather than flexible textile materials, often report longer after-flame times and greater char lengths than those observed for pristine PA6,6 fabrics [37,58,59,60,61]. This further highlights the limitations of direct flame tests in evaluating thermoplastic textiles, particularly when materials demonstrate good performance in other characterization methods.

Scanning electron microscopy (SEM) images of the residue for pristine and treated PA6,6 samples were obtained after the flame test to observe structural changes and residue morphology influenced by the coatings (Figure 14). These images focus on the burned edges surrounding the hole created during the flame test, providing insight into the effects of the treatments on stabilizing the material under flame exposure.

The SEM images reveal a largely similar topography across most samples, with melted PA6,6 fibers merging together, reflecting the polymer’s inherent melting behavior under flame exposure. However, PA6,6 coated with ZnO at a thickness of 199 nm retains the shape of individual fibers to a greater extent, indicating that the ZnO layer provides some structural stabilization during combustion. During the burning process, the underlying PA6,6 fabric melts, leaving behind a continuous ZnO residual layer. For thinner ZnO layers, such as PA6,6-ZnO-84, the residual layer shows signs of cracking and collapse, whereas for thicker ZnO layers (PA6,6-ZnO-199), the layer remains intact, preserving the shape of the original fibers without significant deformation, as shown in Figure 14. The absence of a distinct ZnO layer in the presence of DOPO suggests that a residue of DOPO-based silane forms above the ZnO, potentially integrating with the ZnO layer to create a composite residue, as illustrated in Figure 15.

## 4. Discussion

This study investigated the flame-retardant performance of ZnO coatings alone, the widely studied DOPO-based phosphorus flame retardant, and their combination, applied as dual layers on PA6,6 fabric. The aim was to evaluate their individual and combined effects on the thermal degradation, smoke emission, and combustion behavior of a polymer known for its high flammability and melt-drip characteristics.

ZnO coatings, applied via ALD, were shown to form uniform, conformal layers that maintained the surface structure of the fabric. Advanced structural characterization techniques (SPM, XPS, SAXS/WAXS) confirmed the controlled growth and increasing crystallinity of ZnO with layer thickness. The thermal and combustion analyses (TGA, MCC, cone calorimetry) indicated that ZnO primarily acts as a barrier and catalytic agent, promoting char formation and reducing the release of combustible volatiles.

The DOPO-based silane, applied via sol–gel processing, provided phosphorus functionality capable of modifying the degradation pathway. TG-IR analysis revealed enhanced ammonia release and reduced carbonyl emissions, suggesting altered scission of the amide groups and condensed-phase activity. This behavior is in line with established phosphorus flame-retardant mechanisms. However, despite these changes, DOPO-treated samples did not achieve self-extinguishing performance in direct flame tests, underscoring the limitations of DOPO-based silane treatments alone for polymers like PA6,6.

When applied in combination, the ZnO-DOPO system exhibits additive or partially synergistic effects across multiple analytical techniques. In TGA, the dual-layer coatings result in a decrease in the maximum degradation rate compared to DOPO-treated fabrics, although the rate remains higher than that observed for ZnO-only treated fabrics. An increase in char residue compared to both single-layer systems indicates enhanced condensed-phase stabilization and thermal resistance. MCC further supports this interpretation, showing a pronounced reduction in peak heat release rate (pHRR) and heat release capacity (HRC), along with the formation of an additional decomposition feature, likely associated with the interaction between ZnO and the phosphorus-containing silane. TG-IR analysis shows no further suppression of carbonyl-containing volatiles compared to the DOPO-treated fabric, but still a greater suppression than that achieved by ZnO alone, suggesting a modified degradation pathway upon introduction of DOPO. Cone calorimetry reveals improved smoke suppression and cleaner combustion compared with PA6,6-DOPO; however, a strong synergistic effect on overall flame retardancy is not evident. This may be attributed to the layered architecture, in which the ZnO underlayer limits direct interaction between the DOPO-based silane and the polymer substrate. Notably, the silane network may also act as an additional physical barrier, contributing to reduced volatile release. DOPO-ETES appears to act predominantly in the condensed phase when combined with ZnO.

These results emphasize that while ZnO and DOPO each contribute distinct flame-retardant functions, barrier/catalytic and condensed-phase chemical modification, respectively, their effectiveness in combination depends strongly on coating sequence and interfacial accessibility. While the current configuration, in which the DOPO-based silane is deposited over the ZnO layer, demonstrated some additive or partially synergistic effects, it may also limit the direct interaction of the flame retardant with the polymer surface. This configuration could reduce the potential of the DOPO moiety to influence decomposition pathways at the interface, as suggested by TG-IR data, which showed only partial modification of the volatile profile compared to DOPO-only treatments. Reversing the layer sequence might enhance polymer–DOPO interaction and reveal different flame-retardant behavior. However, such a configuration may also result in the ZnO top layer acting as a barrier, possibly suppressing any gas-phase activity from the DOPO compound.

In parallel, several studies have reported flame-retardant systems for PA6,6 fabrics based on either bio-based compounds, typically applied through layer-by-layer (LbL) techniques, or inorganic/organosilicon coatings formed via sol–gel chemistry. While some of these systems demonstrated effective flame retardancy at relatively low add-on levels, they generally did not address or achieve smoke suppression. By contrast, the ZnO layer in this work, applied via ALD, yielded a pronounced reduction in smoke release and melt-dripping, even with only moderate reduction in peak heat release rate (pHRR). Our dual-layer system, combining ALD with sol–gel processing, demonstrates clear advantages: complete melt-drip suppression, substantial smoke reduction, and the feasibility of applying conformal, scalable coatings. These features underline the broader potential of our approach in comparison to conventional finishing methods.

Table 5 provides an overview of flame-retardant systems for PA6,6 fabrics, summarizing previous studies alongside the present work.

However, to better distinguish the specific role of ZnO in layered flame-retardant architectures, future work will investigate non-silane DOPO derivatives applied in both sequence configurations. This strategy will avoid the additional physical barrier introduced by silane functionalities, which may limit interfacial interactions even when applied as the first layer. By removing the influence of silane, the respective contributions of ZnO and DOPO to gas-phase and condensed-phase activity can be more clearly resolved. Further research should also address how this approach can be extended to other textile substrates.

## 5. Conclusions

This study explored the application of a dual-layer flame-retardant system combining ALD-deposited ZnO and DOPO-based silane to enhance the flame retardancy of PA6,6 fabrics. ALD demonstrated several advantages for the textile sector, including precise control over layer thickness, uniform coverage, and the ability to preserve the textile’s original structure. The ZnO layer not only served as a thermal barrier but also exhibited catalytic activity, influencing the degradation pathways of PA6,6 and DOPO-based silane. This catalytic effect promoted the formation of char and facilitated thermal stability, contributing to the improved flame retardancy observed. The combination with DOPO-based silane further enhanced these effects, significantly improving thermal stability and reducing heat release rates. TG-IR and cone calorimetry analyses also confirmed the role of ZnO in suppressing the release of smoke and incomplete combustion products while highlighting the limited contribution of DOPO to smoke suppression. However, the study also highlighted limitations, particularly the challenges associated with the melting behavior of PA6,6, which hindered compliance with standard flame tests. Additionally, while ALD offers exceptional control and versatility, it is traditionally limited by slow cycle times and batch processing. However, recent advancements in spatial and roll-to-roll ALD technologies have demonstrated the feasibility of high-throughput coatings on flexible substrates, including textiles. Notably, atmospheric-pressure spatial ALD systems have achieved coating speeds exceeding 0.25 m/min on porous materials, underscoring the potential for industrial-scale processing [68]. When combined with the second layer in our system, applied via sol–gel using well-established industrial techniques such as pad–dry–cure finishing, this dual-layer strategy shows strong potential for scalable implementation, pending further optimization of flame-retardant efficacy.

## Figures and Tables

**Figure 1 materials-18-03195-f001:**
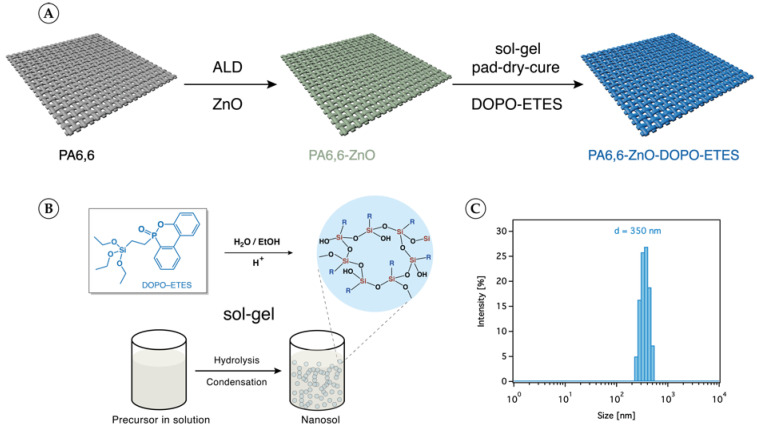
(**A**) A schematic diagram illustrating the dual-layer surface modification of PA6,6 fabric: atomic layer deposition (ALD) of a ZnO underlayer followed by the deposition of a DOPO-based silane coating. (**B**) A schematic representation of the sol–gel synthesis route used to prepare the DOPO-ETES nanosol, involving hydrolysis and condensation reactions of the silane precursor in a water/ethanol mixture under acidic conditions (H^+^). (**C**) The hydrodynamic particle size distribution of the resulting nanosol, measured by dynamic light scattering (DLS) and presented as an intensity-weighted distribution, confirming the formation of uniformly dispersed nanoscale particles.

**Figure 2 materials-18-03195-f002:**
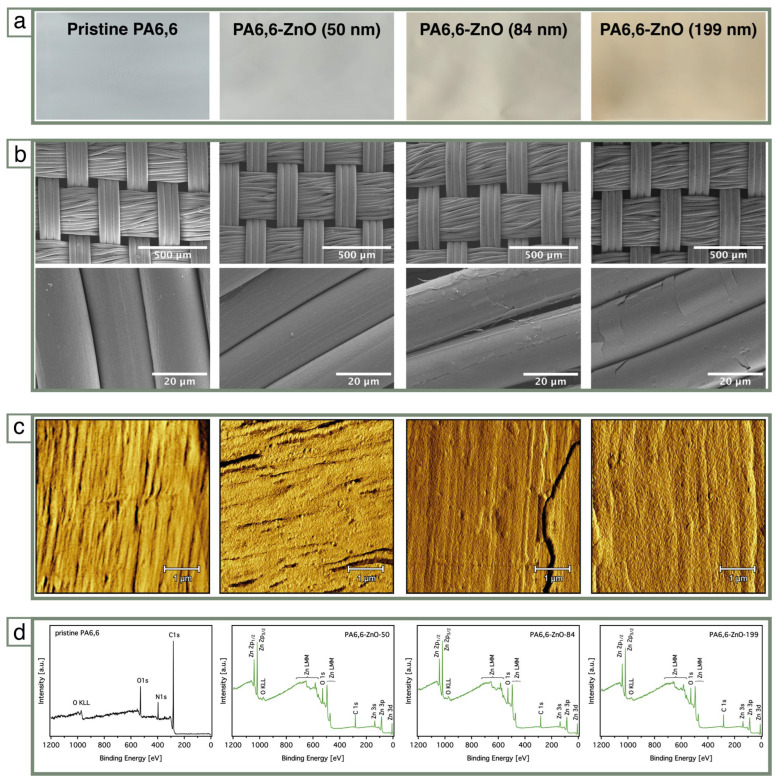
(**a**) Digital photographs of pristine PA6,6 and ZnO-coated fabrics showing the visual appearance with increasing ZnO layer thickness (50, 84, and 199 nm), as deposited by ALD. (**b**) SEM images at low (top row) and high (bottom row) magnification. (**c**) Scanning probe microscopy (SPM) friction images (5 × 5 µm^2^ scan area) reveal nanoscale homogeneity of the ZnO coatings and changes in surface texture compared to pristine PA6,6. (**d**) X-ray photoelectron spectroscopy (XPS) survey spectra of pristine and ZnO-coated fabrics confirming the successful deposition of ZnO.

**Figure 3 materials-18-03195-f003:**
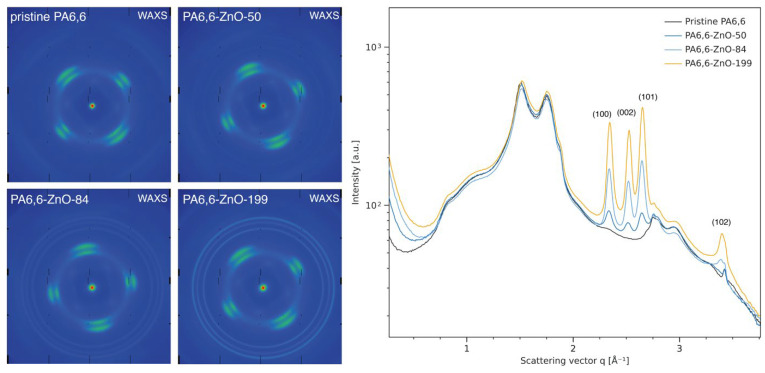
Two-dimensional wide-angle X-ray scattering (WAXS) patterns of pristine and ZnO-coated PA6,6 fabrics (**left**), showing the transition from a diffuse amorphous halo in the uncoated sample to increasingly sharp and well-defined diffraction rings as the ZnO layer thickness increases, indicating the formation of crystalline ZnO domains. The corresponding 1D WAXS line profiles (**right**), obtained by radial integration of the 2D patterns, confirm the emergence of characteristic diffraction peaks assigned to the wurtzite ZnO structure, with increasing intensity and crystallinity at higher deposition thicknesses.

**Figure 4 materials-18-03195-f004:**
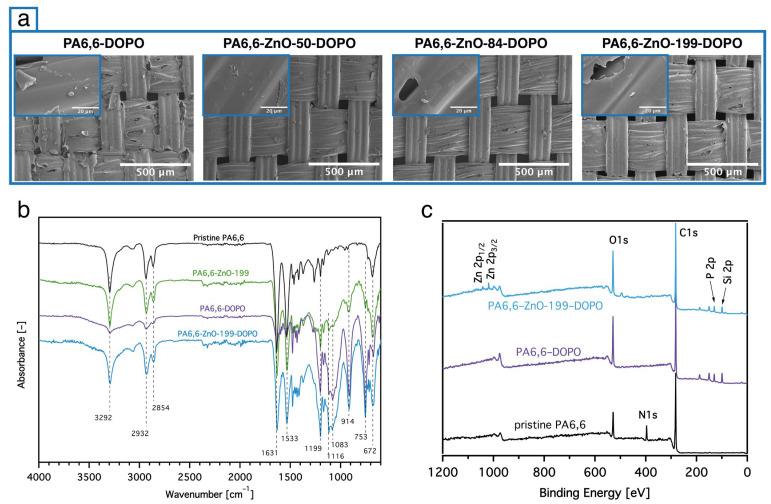
(**a**) SEM images of PA6,6 fabrics coated with DOPO-based silane alone (PA6,6-DOPO) and in combination with ALD-ZnO layers of increasing thickness (50, 84, and 199 nm), showing enhanced surface uniformity in the presence of ZnO. (**b**) ATR-IR spectra of pristine PA6,6, DOPO-coated PA6,6, and ZnO-coated PA6,6 with and without the DOPO-based silane, highlighting the successful incorporation of phosphorus functionalities. (**c**) XPS survey spectra demonstrating the incorporation of Zn, P, and Si elements, verifying the deposition of both ZnO and DOPO-based silane layers on the fabric surface.

**Figure 5 materials-18-03195-f005:**
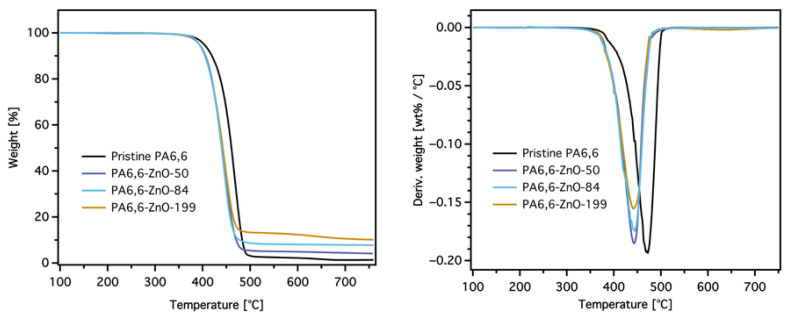
Thermogravimetric analysis (TGA) and the corresponding differential thermogravimetric (DTG) curves of pristine and ZnO-coated PA6,6 fabrics with different layer thicknesses measured under a nitrogen atmosphere at a heating rate of 20 K/min.

**Figure 6 materials-18-03195-f006:**
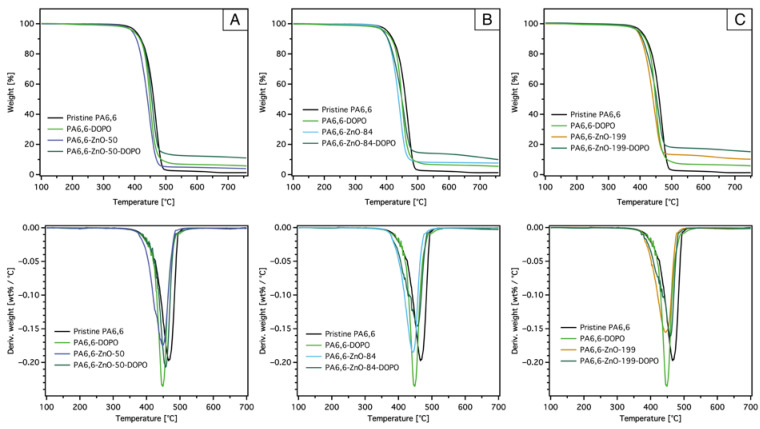
Thermogravimetric analysis (TGA, top) and the corresponding differential thermogravimetric (DTG, bottom) curves of pristine and coated PA6,6 fabrics with DOPO-ETES for different ZnO layer thicknesses, 50 nm (**A**), 84 nm (**B**), and 199 nm (**C**), measured under a nitrogen atmosphere at a heating rate of 20 K/min.

**Figure 7 materials-18-03195-f007:**
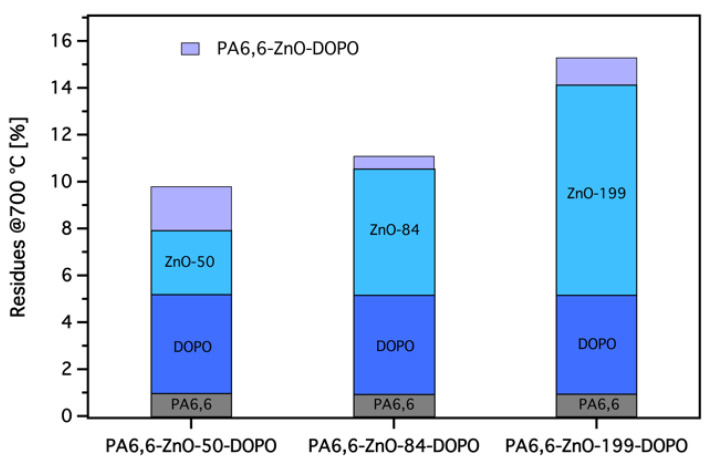
Residual masses of pristine and coated PA6,6 fabrics after TGA testing at 700 °C, highlighting the contribution of each component (PA6,6, ZnO, and DOPO-based silane) to the total residue. The enhanced residue observed for ZnO-DOPO-treated samples exceeds the sum of the individual contributions of ZnO and DOPO when applied separately.

**Figure 8 materials-18-03195-f008:**
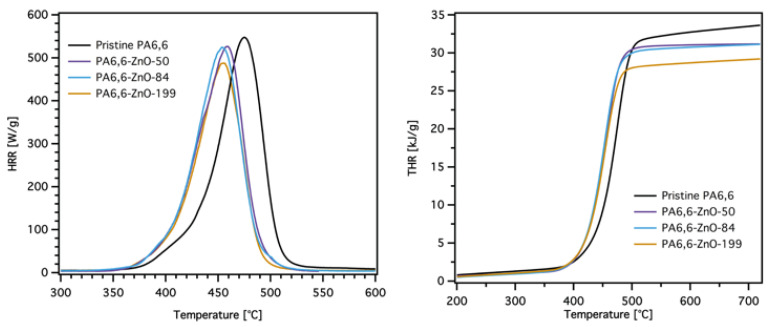
Heat release rate (HRR) and total heat release (THR) curves obtained from microscale combustion calorimetry (MCC) for pristine and ZnO-coated PA6,6 fabrics with varying ZnO layer thicknesses. Increasing the ZnO thickness results in a reduction in both peak HRR and THR.

**Figure 9 materials-18-03195-f009:**
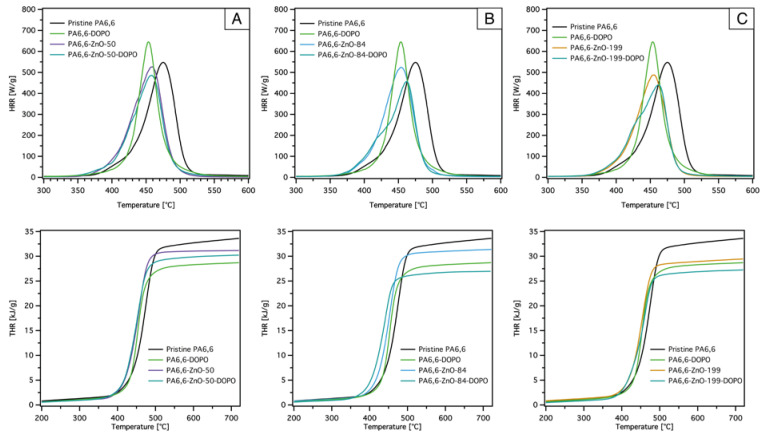
Heat release rate (HRR, top) and total heat release (THR, bottom) curves from microscale combustion calorimetry (MCC) of pristine PA6,6 and PA6,6 fabrics treated with DOPO-based silane, ZnO coatings of different thicknesses, 50 nm (**A**), 84 nm (**B**), and 199 nm (**C**), and their combination. The combined ZnO-DOPO system exhibits a greater decrease in HRR and THR compared to individual treatments.

**Figure 10 materials-18-03195-f010:**
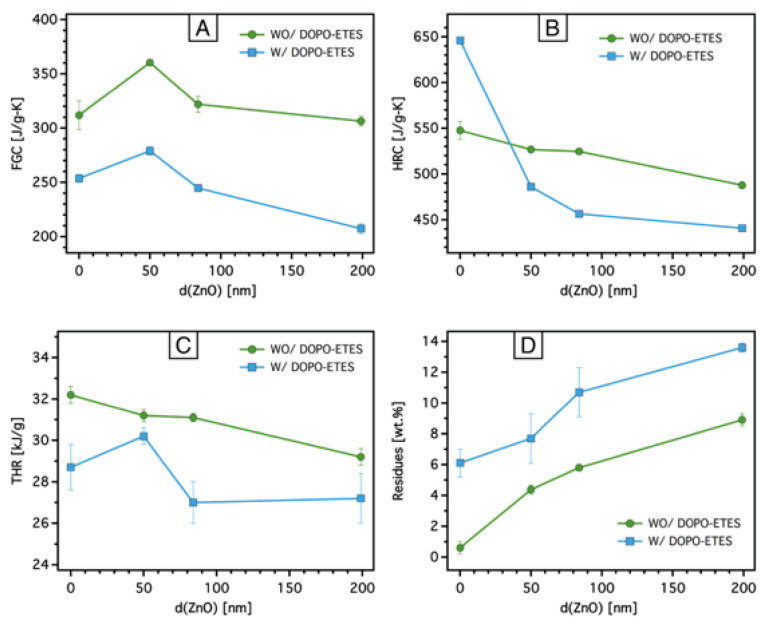
(**A**) Fire growth capacity (FGC), (**B**) heat release capacity (HRC), (**C**) total heat release (THR), and (**D**) residual mass after MCC testing, as a function of ZnO layer thickness on PA6,6 fabrics treated without (wo) and with (w) DOPO-based silane. The combined ZnO-DOPO treatment consistently reduces FGC, HRC, and THR while increasing residue, indicating enhanced flame-retardant performance. Data represent average values from triplicate measurements, and error bars indicate standard deviations (see Table 4).

**Figure 11 materials-18-03195-f011:**
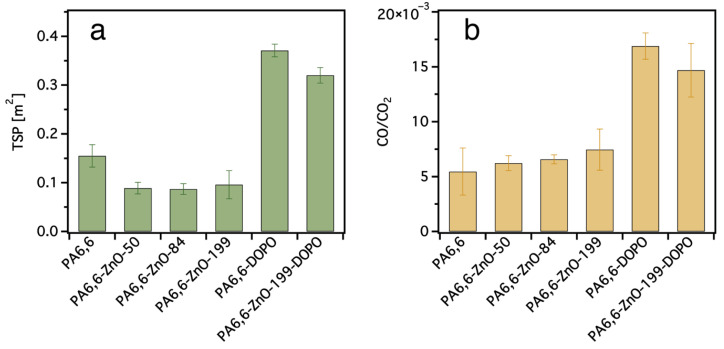
(**a**) The total smoke production (TSP) and (**b**) CO/CO_2_ ratio of pristine and coated PA6,6 samples measured by cone calorimetry. The DOPO-treated sample shows increased TSR and higher CO/CO_2_ ratio, indicating incomplete combustion. The introduction of ZnO reduces both smoke production and CO/CO_2_ ratios, suggesting improved combustion efficiency. The ZnO-DOPO combination results in intermediate values, reflecting partial suppression of smoke and enhanced fire behavior compared to DOPO alone. Data represent average values from triplicate measurements, and error bars indicate standard deviations.

**Figure 12 materials-18-03195-f012:**
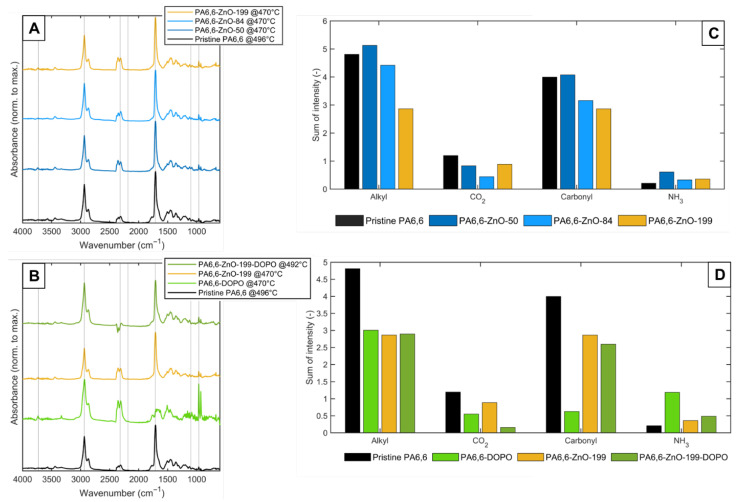
(**A**) IR spectra of evolved gases at T_max_ for PA6,6 and ZnO-coated samples without DOPO. (**B**) Corresponding spectra for samples with DOPO. (**C**) Cumulative emission of typical degradation products (hydrocarbons, CO_2_, carbonyls, and NH_3_) for coatings without DOPO. (**D**) Cumulative emissions for coatings with DOPO.

**Figure 13 materials-18-03195-f013:**
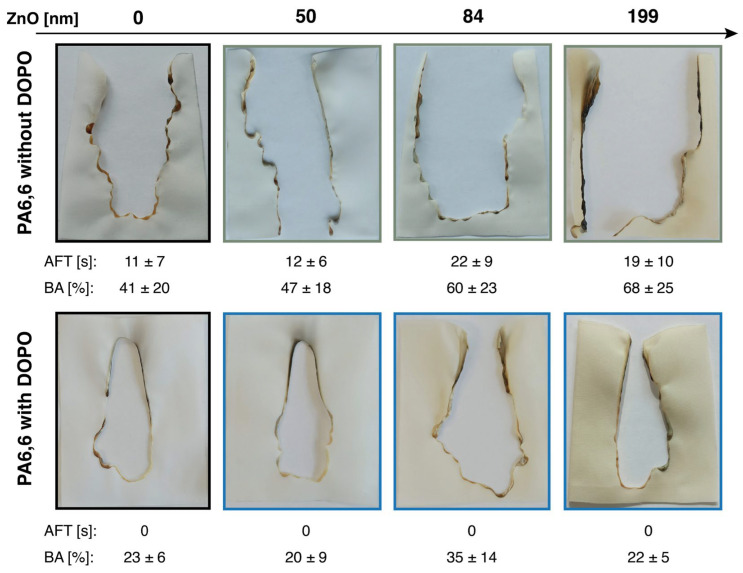
Digital photographs of untreated and treated PA6,6 fabrics with different ZnO layer thicknesses after a flame test using surface ignition, performed according to EN ISO 15025. Modified test specimens with dimensions of 10.5 cm × 6.0 cm were used. Propane was used as the burning gas with a flame height set at 25 mm. The images illustrate the extent of flame damage and material degradation as a function of treatment, with ZnO and DOPO-based coatings. After-flame time (AFT) and burned area (BA, in %) are indicated for each sample, providing semi-quantitative insights into the flame response under the applied conditions. Data represent average values from triplicate testing, and errors indicate standard deviations.

**Figure 14 materials-18-03195-f014:**
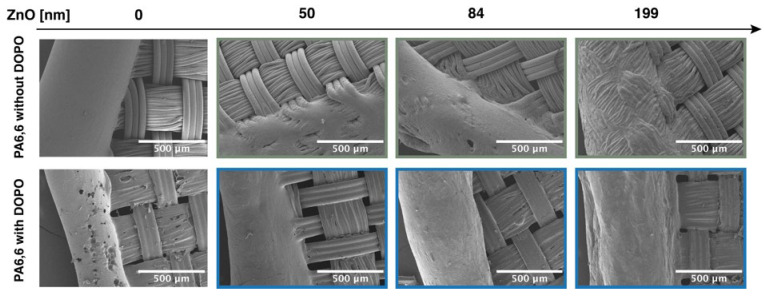
SEM images of the burned edges of pristine PA6,6 and DOPO-treated PA6,6 fabrics with different ZnO layer thicknesses after flame exposure.

**Figure 15 materials-18-03195-f015:**
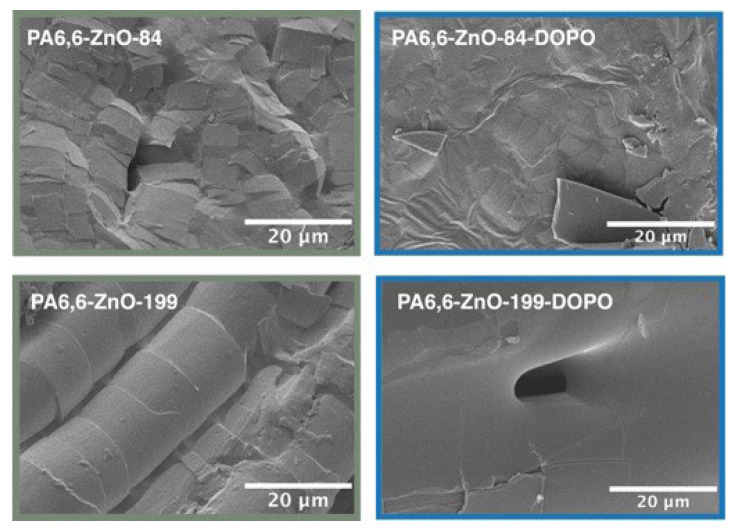
SEM images of burned ZnO-coated PA6,6 fabrics with two different ZnO thicknesses (84 nm and 199 nm), shown with and without DOPO-based silane treatment.

**Table 1 materials-18-03195-t001:** An overview of ZnO atomic layer deposition (ALD) conditions, including the number of deposition cycles, corresponding ZnO layer thickness (measured on reference Si wafers), and the resulting add-on values (wt%) for ZnO-coated PA6,6 fabrics. Data represent average values from five measurements with errors indicating standard deviations (SD).

Sample Code	ZnO		
	Cycles [-]	Layer Thickness [nm]	Add-on [wt%]
PA6,6-ZnO-50	300	50 ± 0.5	3.2 ± 0.5
PA6,6-ZnO-84	600	84 ± 0.8	4.3 ± 0.6
PA6,6-ZnO-199	1200	199 ± 1.9	8.5 ± 0.7

**Table 2 materials-18-03195-t002:** The mass add-on values (weight gain, in wt%) of PA6,6-ZnO fabrics after treatment with DOPO-ETES, along with the corresponding calculated atomic percentages of phosphorus (P) and silicon (Si) and the values obtained from XPS measurements. Except for XPS, all data represent average values from five measurements with errors indicating standard deviations (SD). The SD for the theoretical atomic percentages is based on the SD of the measured add-on values.

Sample Code	DOPO-ETES [wt%]	P_Theor._ [%]	P_XPS_ [%]	Si_Theor._ [%]	Si_XPS_ [%]
PA6,6-DOPO	15 ± 1.4	1.57 ± 0.15	2.3	1.43 ± 0.13	3.1
PA6,6-ZnO-50-DOPO	14.6 ± 1.2	1.53 ± 0.13	1.9	1.39 ± 0.11	1.5
PA6,6-ZnO-84-DOPO	16.1 ± 0.9	1.69 ± 0.09	2.1	1.53 ± 0.09	2.6
PA6,6-ZnO-199-DOPO	14.8 ± 1.1	1.55 ± 0.12	1.6	1.41 ± 0.10	2.8

**Table 3 materials-18-03195-t003:** Thermogravimetric analysis (TGA) data of pristine and coated PA6,6 fabrics measured under a nitrogen atmosphere at a heating rate of 20 K/min.

Sample	T_5%_ [°C]	T_max_ [°C]	Degradation Rate @ T_max_ [wt%/°C] × 10^−2^	Res.700 [%]	Res._Theor_ [%] (PA + ZnO) ^a^	Res._Theor_ [%] (PA + ZnO + SiO_2_) ^b^
PA6,6	401	471	19.3	1.0	–	–
PA6,6-DOPO	391	449	23.5	5.4	–	4.0
PA6,6-ZnO-50	390	443	18.4	3.8	4.2	–
PA6,6-ZnO-50-DOPO	393	457	20.6	9.8	4.2	7.2
PA6,6-ZnO-84	387	449	17.4	6.6	5.3	–
PA6,6-ZnO-84-DOPO	386	457	14.5	11.1	5.3	8.6
PA6,6-ZnO-199	391	445	15.5	10.4	9.5	–
PA6,6-ZnO-199-DOPO	391	459	16.1	15.3	9.5	12.5

^a^: calculated based on the PA6,6 residue at 700 °C and the weight percentage of ZnO derived from Table 1. ^b^: calculated as the sum of (a) and the theoretical weight percentage of SiO_2_ derived from Table 2.

**Table 4 materials-18-03195-t004:** Combustion parameters of pristine and treated PA6,6 fabrics obtained from microscale combustion calorimetry (MCC). Reported values include peak heat release rate (pHRR), temperature at peak HRR (TpHRR), fire growth capacity (FGC), total heat release (THR), and residue at the end of the test. All values represent the average of three independent measurements with corresponding standard deviations. Data represent average values from triplicate measurements, and errors indicate standard deviations.

Sample	pHRR [W/g]	TpHRR [°C]	FGC [J/g-K]	THR [kJ/g]	Residues [%]
PA6,6	547.5 ± 9.9	475.2 ± 1.3	311.7 ± 13.3	33.2 ± 0.4	0.6 ± 0.3
PA6,6-DOPO	646.1 ± 2.9	453.7 ± 0.5	253.6 ± 3.6	28.7 ± 1.1	6.1 ± 0.9
PA6,6-ZnO-50	526.7 ± 2.5	459.4 ± 0.5	360.4 ± 2.7	31.2 ± 0.3	4.38 ± 1.2
PA6,6-ZnO-50-DOPO	485.9 ± 1.2	458.0 ± 0.1	278.9 ± 3.7	30.2 ± 0.4	7.7 ± 1.6
PA6,6-ZnO-84	524.6 ± 2.9	453.9 ± 0	321.9 ± 7.4	31.1 ± 0.2	5.8 ± 0.2
PA6,6-ZnO-84-DOPO	456.5 ± 3.3	461.8 ± 0.4	244.7 ± 0.8	27.0 ± 1.0	10.7 ± 1.6
PA6,6-ZnO-199	487.7 ± 0.7	455.1 ± 0.8	306.3 ± 4.7	29.2 ± 0.4	8.9 ± 0.6
PA6,6-ZnO-199-DOPO	440.8 ± 1.3	447.1 ± 4.1	207.3 ± 4.5	27.2 ± 1.2	13.6 ± 0.3

**Table 5 materials-18-03195-t005:** A summary of flame-retardant systems for PA6,6 fabrics from previous studies and the present work, comparing flame retardant types, finishing methods, FR content, flame retardancy metrics (LOI, pHRR reduction), flame test results, smoke behavior, and dripping tendencies. All abbreviations are defined below the table.

No.	FR Type	Finishing Method	FR Content [wt%]	LOI [%]	Reduction in pHRR [%]	Flame Test	Dripping	Smoke Behavior [%]	Ref.
1	PCS + PAS	LbL + UV/thermal	6.5	23 ± 0.5	25 (CC)	UL 94 (V-1) AFT = 50 ± 8	no	SPR decreased @ FR content of 2.1%	[60]
2	CS/PA/OSA	LbL	9.3	20.7	23 (CC)	UL 94 (V-1) AFT = 18 s	no	N/A	[61]
3	AA/CS/ME/ UREA/PA + CS/GO	UV-grafting + LbL + pad–dry–cure	15.7	25	44.3 (MCC)	UL 94 (V-1) AFT = 33 s	no	N/A	[59]
4	CS/PAA	UV-grafting + pad–dry–cure	30.1	24	52.3 (MCC)	UL 94 (V-1) AFT = 24 s	no	N/A	[58]
5	soybean protein + thiourea	pad–dry–cure	9.4	25 ± 0.4	12.8 (CC)	vertical test ISO 6940-2004 [62] AFT = 0 s	no	N/A	[63]
6	GO-lignin/CS/PA	pad–dry–cure	6.0 ± 0.30	25.5 ± 1	25 (CC)	vertical test ASTMD 6413-08 [64] AFT = 66 ± 13 s	yes	TSR increased by 17.6%	[65]
7	intumescent FR (APP, ME, PER)	pad–dry–cure	40.6	27.9	N/A	vertical test GB/T 5455-1997 [66] AFT = 5.3 s	no	N/A	[67]
8	DOPO-APTES	sol–gel (pad–dry–cure)	14.6	21.5	36 (CC)	UL-94 (V-1) AFT = 38 s	no	SPR increased by 38%	[37]
9	DOPO-DAAM	UV-grafting	20.3	23.0	22 (CC)	UL-94 (V-0) AFT = 10 s	no	N/A	[36]
10	ZnO	ALD	8.5 ± 0.7	N/A	10.9 ± 1.2 (MCC)	EN ISO 15025 (surface ignition) AFT = 19 ± 10 s	no	TSP decreased by 44 ± 8 (vs. pristine PA6,6)	This work
	ZnO + DOPO-ETES	ALD + sol–gel (pad–dry–cure)	23.3 ± 1.8	N/A	19.5 ± 1.6 (MCC)	EN ISO 15025 (surface ignition) AFT = 0 s	no	TSP decreased by 14 ± 1 (vs. PA6,6-DOPO)	This work

**LBL**: layer-by-layer; **CS**: chitosan; **PCS**: phosphorylated chitosan; **PAS**: poly-acrylate sodium; **PA**: phytic acid; **OSA**: oxidized sodium alginate; **ME**: melamine; **GO**: graphene oxide; **AA**: acrylic acid; **PAA**: phytic acid ammonia; **APP**: ammonium polyphosphate; **PER**: pentaerythritol; **APTES**: aminopropyl triethoxysilane; **DAAM**: diallyl amine; **ETES**: ethyl triethoxysilane. **pHRR**: peak heat release rate; **ALD**: atomic layer deposition; **CC**: cone calorimeter test, **MCC**: microscale combustion calorimeter test: **TSP**: total smoke production (m^2^); **TSR**: total smoke release (m^2^/m^2^); **SPR**: smoke production rate (m^2^/s).

## Data Availability

The original contributions presented in this study are included in the article/Appendix A. Further inquiries can be directed to the corresponding authors.

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
