# Peer review of "Thermal Behavior and Smoke Suppression of Polyamide 6,6 Fabric Treated with ALD-ZnO and DOPO-Based Silane"

_materials, 2025, doi:10.3390/ma18133195_

Round 1
Reviewer 1 Report
Comments and Suggestions for Authors
I recommend that this paper be accepted after minor revision.
After carefully reviewing this work, I found that the manuscript has good strength and has shown interesting work with highly appreciated protocol. Still, it requires some minor revisions and the questions below are clarified:
- Showing Data: Error Analysis: The figures don't include error bars or other statistical proof, such as standard deviations from repeated studies. Adding these would make the conclusions more trustworthy. Figure Captions: Add more information to captions to make important points clearer (for example, show how SAXS patterns change between clean and coated textiles in Figure S1). For Figure S3, make it clear how the ZnO-DOPO combo works better than either therapy on its own.
- Details of the Experiment: ALD Parameters: Make sure that all the specifics of the ALD process, such as the number of cycles, kinds of precursors, and temperature, are thoroughly recorded so that they may be repeated. Sample Nomenclature: To prevent confusion, explain what designations like "ZnO-199" mean in the main text or extra material.
- Insights on how things work: Explain in further detail how ZnO and DOPO work together to put out fires. For instance, does ZnO operate as a physical barrier and DOPO stop radical reactions? Talk about how changes in structure seen in SAXS/WAXS (such as crystallinity and nanostructure) are related to heat stability and the capacity to keep things from evaporating.
- Putting things in context: To show how far we've come, compare how well ALD-ZnO/DOPO works with other flame-retardant systems, such as regular coatings or additives. If there are any unexpected outcomes, such as TG-IR profiles that don't follow a straight line, deal with them.

Author Response
# Reviewer 1
I recommend that this paper be accepted after minor revision.
After carefully reviewing this work, I found that the manuscript has good strength and has shown interesting work with highly appreciated protocol. Still, it requires some minor revisions and the questions below are clarified:
Comments 1.1: Showing Data: Error Analysis: The figures don't include error bars or other statistical proof, such as standard deviations from repeated studies. Adding these would make the conclusions more trustworthy.
Response 1.1: We confirm that standard deviations and error bars are provided for all measurements where multiple replicates were performed. Specifically, standard deviations are included for the ZnO-ALD coating thickness and add-on values (Table 1), for the DOPO-based silane mass add-on (Table 2), all MCC data, and for combustion performance parameters such as total smoke production (Figure 10). For measurements such as TGA, only one measurement was conducted per sample, following common practice, as these techniques are known for their high reproducibility and stability under controlled conditions.
To clarify this aspect, we have revised the Materials and Methods section to explicitly indicate where repeated measurements were performed and where single measurements are standard.
Comments 1.2: Figure Captions: Add more information to captions to make important points clearer (for example, show how SAXS patterns change between clean and coated textiles in Figure S1). For Figure S3, make it clear how the ZnO-DOPO combo works better than either therapy on its own
Response 1.2: We thank the reviewer for this suggestion. In response, we have carefully revised and expanded the figure captions throughout the manuscript to improve clarity and highlight the key findings. This includes not only Figures S1 and S3 in the Supporting Information, but also all other figures in the main text and supplementary material.
Comments 2.1: Details of the Experiment: ALD Parameters: Make sure that all the specifics of the ALD process, such as the number of cycles, kinds of precursors, and temperature, are thoroughly recorded so that they may be repeated.
Response 2.1: As indicated in Section 2.2 and summarized in Table 1 of the submitted manuscript, all relevant ALD processing parameters, including the deposition temperature (100 °C), precursor chemicals (diethylzinc and H2O), pulse and purge sequence and timing, carrier gas conditions, and the number of cycles used for each sample, have been provided in full detail. This information ensures that the ZnO ALD process can be accurately reproduced.
Comments 2.2: Sample Nomenclature: To prevent confusion, explain what designations like "ZnO-199" mean in the main text or extra material.
Response 2.2: We thank the reviewer for this helpful suggestion. In response, we have clarified the sample nomenclature in the revised manuscript. Specifically, we now explain that designations such as “ZnO-199” refer to the approximate ZnO layer thickness in nanometers, deposited by ALD using a defined number of cycles. This clarification was added to the revised version to enhance clarity and avoid confusion for the reader.
Comments 3.1: Insights on how things work: Explain in further detail how ZnO and DOPO work together to put out fires. For instance, does ZnO operate as a physical barrier and DOPO stop radical reactions?
Response 3.1: We appreciate the reviewer’s question regarding the mechanism of flame retardancy in the ZnO-DOPO system. While individual mechanisms of ZnO (physical barrier) and DOPO (condensed-phase charring and possible gas-phase activity) are addressed throughout the manuscript, we agree that a clearer summary of their combined effects would enhance clarity. We have therefore added a paragraph to the discussion section summarizing how these two components act together based on the full set of thermal, combustion, and spectroscopic data.
Comments 3.2: Talk about how changes in structure seen in SAXS/WAXS (such as crystallinity and nanostructure) are related to heat stability and the capacity to keep things from evaporating.
Response 3.2: As noted in the manuscript, WAXS and SAXS measurements were primarily conducted to confirm the successful deposition and structural evolution of ZnO layers on the PA6,6 fabrics. No significant structural changes in the PA6,6 substrate were observed following the coating process. Instead, the additional diffraction features detected in both techniques are attributed to the crystalline ZnO layer deposited via ALD, with signal intensity increasing proportionally with layer thickness. These peaks confirm the formation and growth of ZnO nanostructures on the fabric surface. A clarification has been added to the revised version, explicitly stating that no shift or alteration in the diffraction peaks associated with PA6,6 was observed.
Comments 4.1: Putting things in context: To show how far we've come, compare how well ALD-ZnO/DOPO works with other flame-retardant systems, such as regular coatings or additives.
Response 4.1: We appreciate the reviewer’s suggestion. To address this, we have included a comparative summary (Table 5) of previous flame-retardant systems for PA6,6 alongside our current work. While some literature examples report higher LOI values or pHRR reductions at similar or higher FR loadings, our approach, based on the novel combination of atomic layer deposition (ALD) and sol-gel processing, demonstrates clear advantages. These include the complete suppression of melt dripping, notable smoke reduction, and the potential for scalable, conformal coatings. These benefits highlight the relevance of our dual-layer strategy beyond conventional finishing techniques.
Comments 4.2: If there are any unexpected outcomes, such as TG-IR profiles that don't follow a straight line, deal with them.
Response 4.2: We thank the reviewer for this remark. The relevant section has been revised to address
Reviewer 2 Report
Comments and Suggestions for Authors
Limiting the intensity of smoke emission is one of the key aspects related to limiting the flammability of plastics. This is particularly important in the tactics used in seats in public transport, especially railways.
The authors have attempted to limit the flammability of polyamide fabric.
1. The introduction and preface are properly organized and provide a good background for the intended purpose.
2. The literature is a bit sparse. It is worth paying attention to the methods of assessing smoke parameters, various measurement methods in the description.
Garbarski, J., & FabijaÅ„ski, M. (2022). Limitation of smoke emission properties of plastics. Polymers, 49(4), 283–286. Retrieved from https://polimery.ichp.vot.pl/index.php/p/article/view/1752
3. The methodology adopted in the work is correct. Referring to flammability, it is also worth supplementing the research with oxygen index determinations.
4. Very interesting results were obtained from the cone calorimeter research. The authors explained the results in an exhaustive way. Figures and tables are a good supplement.
5. The final conclusions are correctly formulated.
Author Response
# Reviewer 2
Comments and Suggestions for Authors
Limiting the intensity of smoke emission is one of the key aspects related to limiting the flammability of plastics. This is particularly important in the tactics used in seats in public transport, especially railways.
The authors have attempted to limit the flammability of polyamide fabric.
Comment 1: The introduction and preface are properly organized and provide a good background for the intended purpose.
Response 1: We thank the reviewer for the positive feedback.
Comment 2: The literature is a bit sparse. It is worth paying attention to the methods of assessing smoke parameters, various measurement methods in the description.
Garbarski, J., & FabijaÅ„ski, M. (2022). Limitation of smoke emission properties of plastics. Polymers, 49(4), 283–286. Retrieved from https://polimery.ichp.vot.pl/index.php/p/article/view/1752
Response 2: We thank the reviewer for this suggestion and for sharing the reference. Unfortunately, the article by Garbarski and Fabijański (2022) is published in Polish, and no full English translation was available to allow for a thorough evaluation of its methodological details. As indicated in the abstract, the study primarily references national Polish standards for assessing smoke emission, which may differ from international or widely adopted protocols such as ISO or ASTM standards. Therefore, its integration into our comparative discussion was limited by accessibility and methodological relevance.
In response to the broader comment regarding the sparsity of literature, we have now expanded the introductory and discussion sections to incorporate more references related to smoke suppression mechanisms and measurement techniques in flame-retardant research. Furthermore, we have clarified the experimental methodologies used in this study, including cone calorimetry, which is widely accepted standard for evaluating combustion and smoke parameters in textiles.
Comment 3: The methodology adopted in the work is correct. Referring to flammability, it is also worth supplementing the research with oxygen index determinations.
Response 3: We thank the reviewer for this suggestion. In addition to the reported flammability tests, we also attempted to measure the LOI to further evaluate the flame retardant performance of the treated fabrics. However, due to the melting and shrinking behavior of the PA6,6 textile, it was not possible to determine accurate LOI values for either the pristine or the treated samples. This limitation is commonly encountered with thermoplastic fabrics, where melt-dripping and deformation compromise the reliability of LOI measurements. Nevertheless, we have added a corresponding statement to the revised manuscript to clarify this point.
Comment 4: Very interesting results were obtained from the cone calorimeter research. The authors explained the results in an exhaustive way. Figures and tables are a good supplement.
Response 4: As noted in the manuscript, due to the limited available sample mass (~1 g), the cone calorimeter tests deviated from standard protocols, which typically require 20–50 g of specimen to ensure stable combustion and reproducible heat release data. As such, selected quantitative parameters (e.g., pHRR, THR, TTI, and MARHE) exhibited inconsistent and non-linear trends. Notably, the pristine PA6,6 sample occasionally showed lower peak values than the treated ones, contrary to expectations. Therefore, these results were not discussed in detail in the main manuscript.
For transparency and completeness, all corresponding cone calorimetry figures and data, including those not discussed in the main text, have been included in the Supporting Information and we have added a corresponding statement to the revised manuscript. I addition we revised the part related to smoke suppression.
Comment 5: The final conclusions are correctly formulated.
Response 5: We thank the reviewer for the positive feedback.
Reviewer 3 Report
Comments and Suggestions for Authors
Below is my review of the manuscript “Thermal behavior and smoke suppression of polyamide 6,6 fabric treated with ALD-ZnO and DOPO-based silane.” I’ve tried to balance praise for what works well with polite, constructive questions and suggestions, referring to line numbers where it may help. The authors present a dual-layer approach—atomic layer deposition (ALD) of ZnO beneath a DOPO-ETES silane sol–gel coating—to improve flame retardancy and smoke suppression of PA6,6 fabrics (lines 17–24). They show that ZnO catalyzes char formation and that the combination with DOPO-ETES reduces heat release and smoke-related volatiles, though self-extinguishing was not achieved (lines 21–29). The work is thorough in characterization (TGA, MCC, TG-IR, cone calorimetry), and the idea of combining a catalytic inorganic layer with a phosphorus silane is interesting.
Major Comments and Questions
- Mechanistic insight into DOPO activity (TG-IR): You report no clear P-containing volatiles in TG-IR at Tmax (lines 638–642). Could you discuss whether potential overlaps with PA6,6 signals might hide gaseous phosphorus species? Have you considered complementary techniques (e.g. Py-GC/MS) to capture any low-concentration P-volatiles that TG-IR might miss?
- Layer sequence and interfacial access: In section 3.4 you note limited interaction of DOPO when deposited over ZnO (lines 740–748). Have you considered reversing the sequence (silane first, then ALD) or co-depositing both simultaneously to improve synergy? A brief discussion or preliminary data would strengthen your conclusions.
- Flame-test reproducibility and quantification: You mention difficulty in reproducing direct flame tests (lines 660–668). Could you provide more details on the variability? For example, what was the range of after-flame times or burned-area percentages across repeats? Even if self-extinguishing was not achieved, could you include quantitative metrics (e.g., time to flameout, char length) to aid comparison?
- Statistical treatment of measurements: For ALD thickness (±0.5 nm etc.), TGA/MCC (triplicate averages), and other data, please clearly state the number of repeats and report standard deviations or confidence intervals in each table and figure caption.
- Scalability and practical relevance: ALD can be slow and costly for textiles. Could you comment in the Conclusions (or a dedicated section) on the feasibility of scaling this dual-layer approach to industrial fabric widths and throughputs?
- Line 50–55: When introducing MOFs as flame retardants, please cite the very recent 2024 reviews by Qiu et al. and Song et al.
- Line 115–116: You hypothesize that ZnO underlayer will bond strongly with DOPO-ETES. Can you provide any XPS high-resolution spectra (e.g. Si 2p, P 2p) to support direct chemical bonding versus simple adsorption?
- Table 2 (line 333): You report ~15 wt% add-on by DOPO-ETES. It would help to list the corresponding theoretical P and Si contents alongside measured XPS atomic percentages, to confirm coating uniformity.
- Figure 13 caption (line 674): Indicate the exact sample dimensions and propane flow rate used, for reproducibility.
Author Response
# Reviewer 3
Comments and Suggestions for Authors
Below is my review of the manuscript “Thermal behavior and smoke suppression of polyamide 6,6 fabric treated with ALD-ZnO and DOPO-based silane.” I’ve tried to balance praise for what works well with polite, constructive questions and suggestions, referring to line numbers where it may help. The authors present a dual-layer approach—atomic layer deposition (ALD) of ZnO beneath a DOPO-ETES silane sol–gel coating—to improve flame retardancy and smoke suppression of PA6,6 fabrics (lines 17–24). They show that ZnO catalyzes char formation and that the combination with DOPO-ETES reduces heat release and smoke-related volatiles, though self-extinguishing was not achieved (lines 21–29). The work is thorough in characterization (TGA, MCC, TG-IR, cone calorimetry), and the idea of combining a catalytic inorganic layer with a phosphorus silane is interesting.
Major Comments and Questions
Comment 1: Mechanistic insight into DOPO activity (TG-IR): You report no clear P-containing volatiles in TG-IR at Tmax (lines 638–642). Could you discuss whether potential overlaps with PA6,6 signals might hide gaseous phosphorus species? Have you considered complementary techniques (e.g. Py-GC/MS) to capture any low-concentration P-volatiles that TG-IR might miss?
Response 1: We thank the reviewer for this valuable comment. As stated in the manuscript, no distinct IR absorption bands corresponding to gaseous phosphorus-containing species were observed in the TG-IR spectra at the maximum degradation temperature (Tmax). Typically, P–O and P=O stretching vibrations of P-containing species are expected in the region between 900-1250 cm–1. However, in our samples, this region overlaps significantly with strong absorption bands from PA6,6 degradation products, particularly those arising from C–O, C–N, and N–H vibrations, which can mask weak phosphorus-related signals. This spectral congestion could obscure the detection of low-intensity phosphorus-containing volatiles.
Regarding complementary techniques such as Py-GC/MS, we acknowledge its usefulness in identifying trace gaseous species. However, due to limited access to such instrumentation, we were unable to perform these measurements within the scope of this study. Nonetheless, previous reports suggest that when DOPO is functionalized with silane groups, as in the DOPO-ETES used here, the flame retardant mechanism shifts more strongly toward condensed-phase activity for PA6,6 fabrics. In contrast, DOPO derivatives without silane functionality are more likely to exhibit more gas-phase activity. The absence of clear phosphorus volatiles in our TG-IR results is therefore consistent with a condensed-phase dominant mechanism.
A corresponding statement has been added to the manuscript to address this point.
Comment 2: Layer sequence and interfacial access: In section 3.4 you note limited interaction of DOPO when deposited over ZnO (lines 740–748). Have you considered reversing the sequence (silane first, then ALD)
A brief discussion or preliminary data would strengthen your conclusions.
Response 2: We thank the reviewer for this suggestion. In our current layered configuration, the DOPO-based silane is deposited over the ZnO layer, which likely limits its direct interaction with the PA6,6 substrate, as suggested by our TG-IR data. This reduced interfacial access may constrain its flame-retardant efficiency. While reversing the sequence, applying the DOPO-based silane first, followed by ZnO deposition via ALD, could potentially improve polymer interaction, it might also result in the ZnO top layer acting as a barrier, suppressing any gas-phase flame inhibition mechanisms associated with DOPO. In addition, as we discussed in the previous comment, previous reports suggest that when DOPO is functionalized with silane groups, the flame retardant mechanism shifts more strongly toward condensed-phase activity.
To better distinguish the specific role of ZnO in layered flame-retardant architectures, future work will investigate non-silane DOPO derivatives applied in both sequence configurations. This strategy will avoid the additional physical barrier introduced by silane functionalities, which may limit interfacial interactions even when applied as the first layer. By removing the influence of silane, the respective contributions of ZnO and DOPO to gas-phase and condensed-phase activity can be more clearly resolved.
A corresponding statement has been added to the revised manuscript in the Discussion section.
Comment 3: Flame-test reproducibility and quantification: You mention difficulty in reproducing direct flame tests (lines 660–668). Could you provide more details on the variability? For example, what was the range of after-flame times or burned-area percentages across repeats? Even if self-extinguishing was not achieved, could you include quantitative metrics (e.g., time to flameout, char length) to aid comparison?
Reponse 3: In response, the revised version of the manuscript now includes additional details and discussion on the observed variability in the direct flame tests, specifically the range of after-flame times and burned-area percentages across repeated measurements. Other quantitative metrics such as char length could not be reliably determined, as most samples did not produce char. Additional information also added to the related characterization part in the measurements and characterization section.
Comment 4: Statistical treatment of measurements: For ALD thickness (±0.5 nm etc.), TGA/MCC (triplicate averages), and other data, please clearly state the number of repeats and report standard deviations or confidence intervals in each table and figure caption.
Response 4: We confirm that standard deviations and error bars are provided for all measurements where multiple replicates were performed. Specifically, standard deviations are included for the ZnO-ALD coating thickness and add-on values (Table 1), for the DOPO-based silane mass add-on (Table 2), all MCC data, and for combustion performance parameters such as total smoke production (Figure 10). For measurements such as TGA, only one measurement was conducted per sample, following common practice, as these techniques are known for their high reproducibility and stability under controlled conditions.
To clarify this aspect, we have revised the Materials and Methods section to explicitly indicate where repeated measurements were performed and where single measurements are standard. In addition, standard deviations are now reported in all relevant figure and table captions.
Comment 5: Scalability and practical relevance: ALD can be slow and costly for textiles. Could you comment in the Conclusions (or a dedicated section) on the feasibility of scaling this dual-layer approach to industrial fabric widths and throughputs?
Response 5: We thank the reviewer for this insightful comment. While conventional ALD is indeed limited by slow cycle times and batch processing, recent advancements in spatial ALD and roll-to-roll ALD technologies have demonstrated the feasibility of high-throughput coating on flexible substrates, including textiles. Notably, atmospheric-pressure spatial ALD systems have achieved coating speeds exceeding 0.25 m/min on porous materials, highlighting the potential for industrial-scale processing (see: https://doi.org/10.1116/1.4893428). Additionally, the second layer in our dual system, applied via sol-gel, relies on techniques such as pad-dry-cure or spraying, which are already well established in textile manufacturing. Based on these developments, we believe that the dual-layer approach proposed in our work holds promise for scalable, practical implementation, pending further optimization of flame retardant efficacy.
Comment has been added to the conclusion
Comment 6: Line 50–55: When introducing MOFs as flame retardants, please cite the very recent 2024 reviews by Qiu et al. and Song et al.
Response 6: The mentioned references were included in the revised manuscript
Comment 7: Line 115–116: You hypothesize that ZnO underlayer will bond strongly with DOPO-ETES. Can you provide any XPS high-resolution spectra (e.g. Si 2p, P 2p) to support direct chemical bonding versus simple adsorption?
Response 7: We appreciate the reviewer's comment. Although direct confirmation via high-resolution XPS was not performed in this study, it is important to clarify that the use of silicon dioxide (SiO2) and silane-based derivatives in conjunction with zinc oxide (ZnO) is a well-established and extensively studied area in materials science. Numerous research efforts have focused on synthesizing, characterizing, and applying ZnO/SiO2 composites and silane-modified ZnO for various functionalities.
For instance, the following review articles highlight the widespread research on: ZnO/SiO2 composites and Silane-modified ZnO (https://doi.org/10.1016/j.matchemphys.2016.11.011 ; https://doi.org/10.1016/j.porgcoat.2015.05.023 ; https://doi.org/10.3390/pr11041193
https://doi.org/10.1016/j.apsusc.2010.06.091; https://doi.org/10.3390/nano7120439 ;
https://doi.org/10.1080/07853890.2021.1896916)
Therefore, our approach builds upon this substantial body of existing literature which is also added in the reviesd manuscript.
Comment 8: Table 2 (line 333): You report ~15 wt% add-on by DOPO-ETES. It would help to list the corresponding theoretical P and Si contents alongside measured XPS atomic percentages, to confirm coating uniformity.
Response 8: We have added the corresponding measured XPS atomic percentages in Table 2 and discussed in the text.
Comment 9: Figure 13 caption (line 674): Indicate the exact sample dimensions and propane flow rate used, for reproducibility.
Response 9: The sample dimensions (10.5 cm × 6.0 cm) and the flame height (25 mm) have now been added to the caption of Figure 13 for clarity and reproducibility. Unfortunately, the exact propane flow rate could not be determined with our current setup; however, the burner was adjusted to produce a standardized 25 mm flame height, as specified in EN ISO 15025.
Round 2
Reviewer 3 Report
Comments and Suggestions for Authors
Current version of the manuscript can be recommended for publication.